# Improved YOLOv7-Tiny Complex Environment Citrus Detection Based on Lightweighting

Bo Gu [1], Changji Wen [1,2], Xuanzhi Liu [1], Yingjian Hou [2], Yuanhui Hu [1] and Hengqiang Su [1,*]

1 College of Information and Technology, Jilin Agricultural University, Changchun 130118, China; 20211370@mails.jlau.edu.cn (B.G.); changjiw@jlau.edu.cn (C.W.); liuxuanzhi@mails.jlau.edu.cn (X.L.); 20211367@mails.jlau.edu.cn (Y.H.)
2 Institute for the Smart Agriculture, Jilin Agricultural University, Changchun 130118, China; houyingjian@ygemail.com
* Correspondence: suhengqiang@jlau.edu.cn

**Abstract:** In complex citrus orchard environments, light changes, branch shading, and fruit overlapping impact citrus detection accuracy. This paper proposes the citrus detection model YOLO-DCA in complex environments based on the YOLOv7-tiny model. We used depth-separable convolution (DWConv) to replace the ordinary convolution in ELAN, which reduces the number of parameters of the model; we embedded coordinate attention (CA) into the convolution to make it a coordinate attention convolution (CAConv) to replace the ordinary convolution of the neck network convolution; and we used a dynamic detection head to replace the original detection head. We trained and evaluated the test model using a homemade citrus dataset. The model size is 4.5 MB, the number of parameters is 2.1 M, mAP is 96.98%, and the detection time of a single image is 5.9 ms, which is higher than in similar models. In the application test, it has a better detection effect on citrus in occlusion, light transformation, and motion change scenes. The model has the advantages of high detection accuracy, small model space occupation, easy application deployment, and strong robustness, which can help citrus-picking robots and improve their intelligence level.

**Keywords:** computer vision; deep learning; attention mechanism; citrus detection; YOLOv7-tiny

## 1. Introduction

Citrus is one of the most critical global cash crops and one of the most productive fruit categories in the world. In South China, citrus is the fruit tree with the most extensive cultivation area and the most critical economic status [1]. In 2022, the national citrus production amounted to 6.0389 million tons [2]. However, due to the complex environment of citrus picking, picking still mainly relies on manual operation, which is the most significant labor input in the fruit production process, accounting for 50% to 70% of the overall workload [3]. To solve this problem, in recent years, the development of intelligent agriculture has promoted the research of fruit-picking robots [4]. The vision system is one of the critical components of picking robots [5], and how to accurately identify and detect citrus fruits in complex environments will directly affect the accuracy and efficiency of the determining robots [6].

Traditional image techniques for fruit detection usually require manual feature extraction, mainly through features such as color [7], shape [8], and texture [9], and feature extraction is highly dependent on the researcher's experience. In addition, the detection accuracy is low, and the real-time generalization ability is poor in the case of changing lighting conditions, the presence of occlusion of the fruit [10], the similarity of the fruit and the background color [11,12], and even weather and environmental changes. In addition, the detection and classification of citrus fruits can also be performed using infrared imaging and multispectral imaging techniques [13–17], but the equipment cost is high and requires expertise.

With the development of deep learning in recent years, deep convolutional neural network (CNN) has become a mainstream algorithm for fruit target detection due to its strong robustness and generalization ability [18]. It has been studied by many scholars [19–22]. CNN-based target detection algorithms can be divided into two stages and one stage. The two-stage detection algorithm first generates candidate regions that may contain the target through several strategies, and these candidate regions are then passed through a convolutional neural network to classify samples to determine if they contain the target object [23]. The representative algorithms are Fast R-CNN [24] and Faster R-CNN [25]. Using Faster R-CNN, Xiong et al. [26] conducted experiments on green mandarin oranges in the natural environment, which were divided into different light, size, and numbers to carry out, and the average detection accuracies were 77.45%, 73.53%, and 82.58%, respectively. Juntao et al. [27] used the Faster R-CNN model to detect green mandarin oranges on the tree, and the average of the test set accuracy (mAP) was 85.49%. The average computation time for seeing a single image was 0.4 s. However, the two-stage detection algorithm suffers from higher complexity and slower detection speed.

The one-stage detection algorithm directly predicts the category probability and location of the output target, which reduces the complexity while increasing the detection speed and thus receives more attention [28]. The representative algorithms are SSD [29] and the YOLO series [30–36]. Li et al. [37] proposed a citrus detection algorithm based on improved SSD, with an average accuracy of 87.89%. However, the dataset of the paper was derived from the laboratory and differed from the natural environment. Lv et al. [38] proposed a citrus recognition method based on improved YOLOv3-LITE with an average precision (AP) value of 92.75%. However, the model occupies a large amount of memory, which could not be more conducive to practical deployment. Mirhaji [39] et al. used YOLOv4 for detecting and counting oranges in an orchard, with an accuracy, recall, F1, and mAP of 91.23%, 92.8%, 92%, and 90.8%, respectively. However, the method needs to improve its average precision. Chen et al. [40] used the improved YOLOv4 citrus detection algorithm and pruned the trained model with an average accuracy of 96.04%. Zheng et al. [41] pruned the backbone of YOLOv4 and removed the redundant portion of the neck network to propose the YOLO BP green citrus detection algorithm with an average precision of 91.55%. Huo et al. [42] used "Shantanju" citrus collected from Conghua, Guangzhou, and improved the YOLOv5s algorithm to detect and locate mature citrus. The recall rates under uneven, weak, and good lighting were 99.55%, 98.47%, and 98.48, respectively. The average detection time per frame was 78.96 ms. However, the detection speed of this method needs to be improved. Xinyang et al. [43] improved YOLOv5s by introducing ShuffleNetV2, SimAM attention, and Alpha-IoU using images taken from Foshan citrus orchards in Guangdong Province, as well as pictures acquired from the web, and proposed the YOLO-DoC citrus detection method, with a $p$ value and a mAP value of 98.8% and 99.1%, respectively, and an FPS of 187. However, the model's training samples need to be increased. In a study by Liu et al. [44], by using CA attention, replacing PAFPN with BiFPN, and using the zoom loss function, YOLOv5 was improved to detect four varieties of citrus collected from Lingui City, Guilin, Guangxi, namely, "Kumquat", "Nanfeng tangerine", "Fertile tangerine", and "Shatang tangerine". "Fertile tangerine" and "Shatang tangerine" were collected from Lingui City, Guilin, Guangxi Province, and the mAP was 98.4% and 98.4%, and the detection time for a single image was 19 ms. However, the number of parameters for this model is large. The above research method supports the application of CNN on citrus detection and provides a reference for designing subsequent algorithms. However, the current citrus detection algorithms need more research on citrus detection work in complex environments. Table 1 provides the citrus variety, dataset sizes, models used, and experimental results for some references.

Firstly, citrus-picking robots face different fruit sizes, random distribution, fruit overlapping, and branch and leaf shading when picking citrus grown in natural environments. Secondly, the light intensity and angle of light in the natural environment vary randomly, which significantly impacts the image quality. Thirdly, the picking robot will cause blurring

of the image due to the varying movement speed when the material is outdoors. Finally, when using models trained by deep convolutional neural networks, they tend to require more significant memory, which could be more conducive to deploying edge devices.

**Table 1.** Some reference datasets, models, and results.

| Authors | Citrus Variety | Number of Images | Image Size | Main Model | mAP0.5 (%) | FPS (GPU) | Params (M) | MS (MB) |
|---|---|---|---|---|---|---|---|---|
| Li et al. [37] | | 2500 | 768 × 768 | SSD | 87.89 | 49.33 | / | / |
| Lv et al. [38] | Citrus | 620 | 416 × 416 | YOLOv3 | 91.13 | 59.17 | / | 28.00 |
| Mirhaji [39] et al. | IranDezful's orange | 766 | 3000 × 4000 and 2592 × 3872 | YOLOv4 | 90.80 | 42.37 | / | / |
| Chen et al. [40] | Kumquats, Nanfeng tangerines | 1750 | 512 × 424 | YOLOv4 | 96.04 | 16.67 | / | 187.00 |
| Zheng et al. [41] | emperor citrus and tangerine citrus | 890 | 4496 × 3000 and 2592 × 1944 and 3042 × 4032 | YOLOv4 | 91.55 | 18.00 | / | / |
| Huo et al. [42] | Shantanju | 4855 | 1920 × 1080 | YOLOv5s | / | 12.66 | / | / |
| Xinyang et al. [43] | Citrus | 1435 | 640 × 480 | YOLOv5s | 99.10 | 187.00 | / | 2.80 |
| Liu et al. [44] | Kumquat, Nanfeng tangerine, Fertile tangerine, and Shatang tangerine | 1500 | 1280 × 720 | YOLOv5l | 98.40 | 52.63 | 50.90 | / |

This paper proposes a lightweight citrus detection model based on YOLOv7-tiny for recognizing ripe fruits in a complex citrus orchard environment. By comparing different light backbone networks, the use of depth separable convolution (DWConv) to replace the regular convolution in an efficient layer aggregation network (ELAN) to reduce the number of parameters in the model is finally determined. In the neck network, the coordinate attention mechanism (CA) is combined with ordinary convolution to form CAConv, which is experimentally shown to improve the feature extraction ability of the model. In the detection part, the Dynamic Head is used instead of the ordinary detection head to improve the model's ability to detect citrus fruits at different scales based on fusing multi-layer features.

The lightweight citrus detection model proposed in this paper can provide vision algorithm support for the picking robot and improve the robot's environmental adaptability. Meanwhile, this model's small memory and low computation make it easy to deploy on the robot. In addition, the method proposed in this paper achieves fast real-time detection of citrus.

## 2. Materials and Methods

### 2.1. Data Acquisition

We collected image data of a ripe mandarin citrus named "Yongxing Bingtang Citrus" in the citrus plantation in Yongxing County, Hunan Province. We used the Mi10Ultra (Xiaomi Technology Co., Ltd., Beijing, China) as the acquisition equipment, and the acquired images had a size of 4000 × 3000 pixels. The collected data comprised citrus images taken at different times of the day under various lighting conditions, sizes of citrus trees, and angles. The distance between the capture device and the citrus ranged from 0.3 m to 2 m. We aligned the images with citrus grown in its natural environment, which included front-light, Dark-light, overlap, and occlusion. After processing, we obtained 1908 images with precise target contours and textures. Figure 1 shows examples of a citrus orchard

in a natural environment with lush foliage and more occluded fruits, creating a complex picking scene.

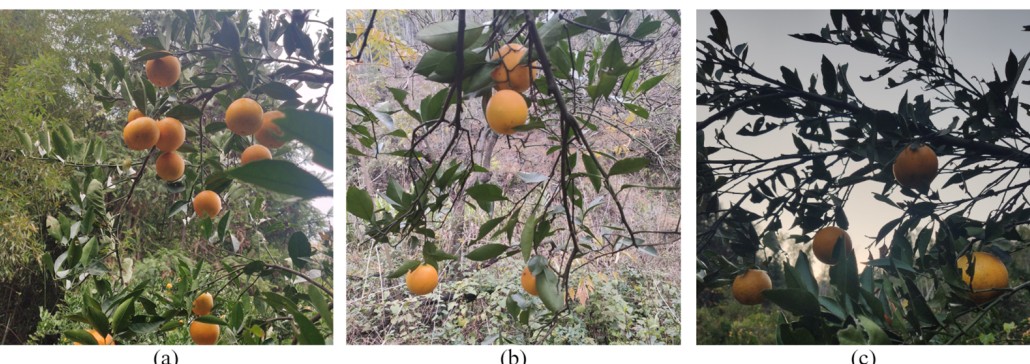

| (a) | (b) | (c) |

**Figure 1.** Collected citrus images from an orchard, including (**a**) a side-lit citrus, (**b**) occluded citrus, and (**c**) a dark-lit citrus.

*2.2. Dataset Construction*

From the above dataset, 120 samples targeting citrus with less than 30% surface shading were selected as mild shading test set A. Part of test set A is shown in Figure 2. In addition, 120 citrus targets targeting citrus with more than 30% surface shading and relatively dense fruit distribution were used as severe shading test set B. Part of test set B is shown in Figure 3. The original complete test set is A + B with 240 images. Then, 1500 pictures were randomly selected from the remaining authentic images as the training set and 168 as the validation set.

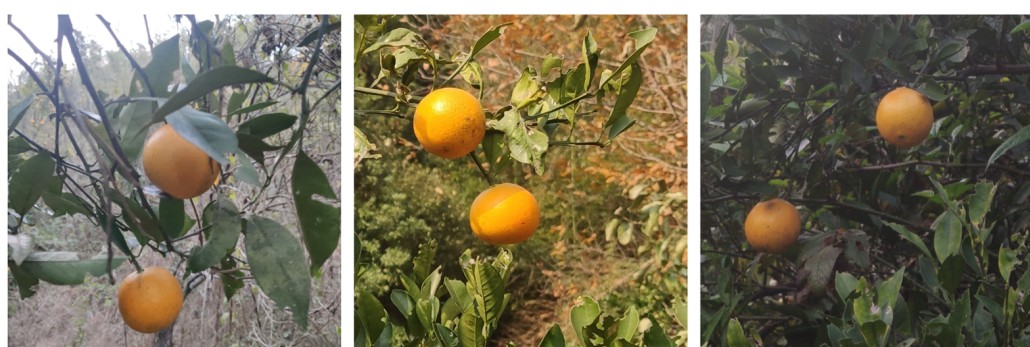

**Figure 2.** Slightly occluded images of citrus in different viewpoints.

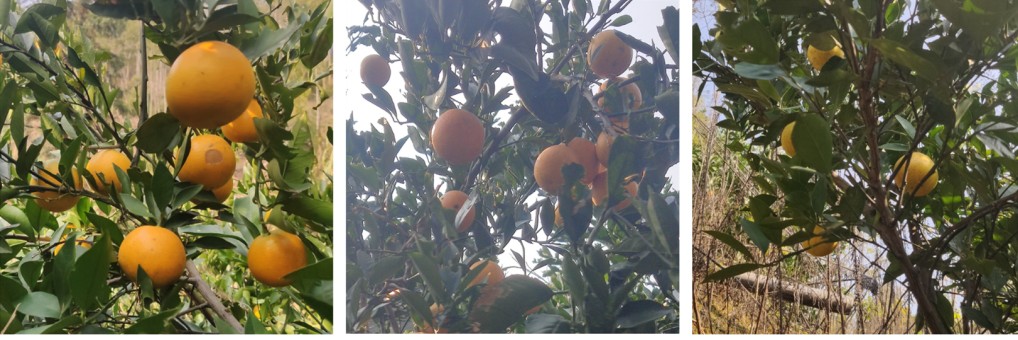

**Figure 3.** Severely occluded images of citrus from different viewpoints.

In the actual data collection, it is difficult to cover all application scenarios, so to enhance the model's robustness, improve the model's generalization ability, and reduce the risk of overfitting the model, a data augmentation strategy was introduced to expand

the dataset. The augmentation techniques used are random brightness variation, $-90°$ to $90°$ random flip, Gaussian noise, motion blur, random erasure, and Mosaic. The data augmentation diagram is shown in Figure 4.

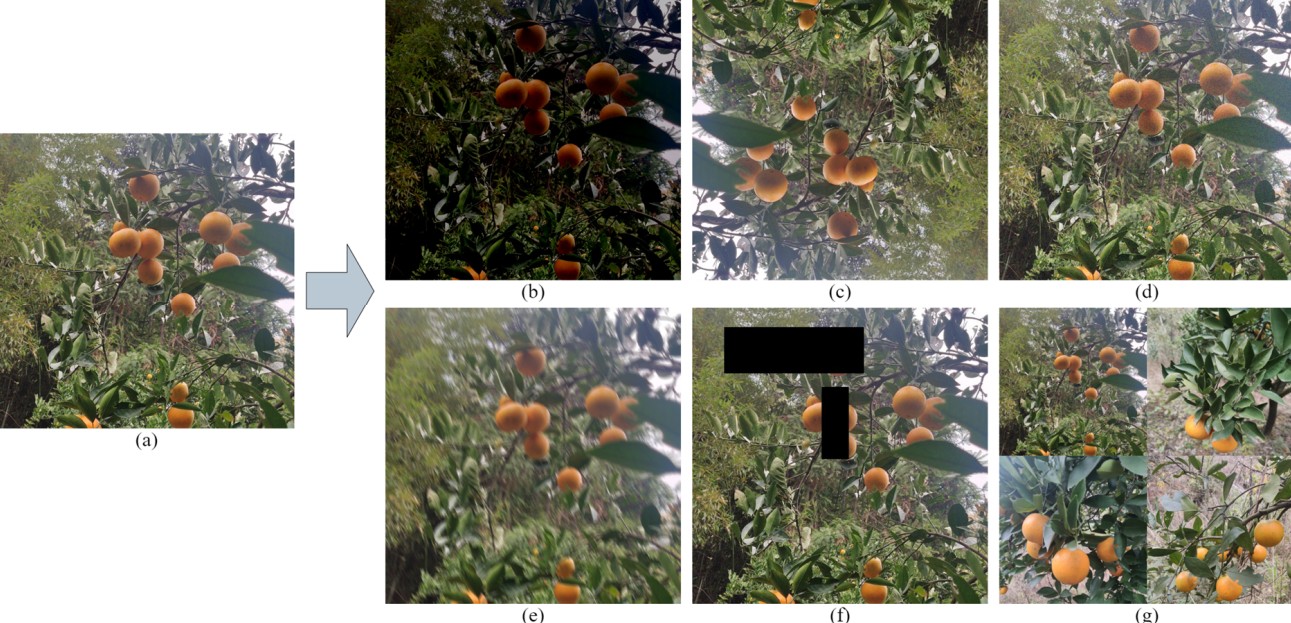

**Figure 4.** Citrus data augmentation images, (**a**) original image, (**b**) random luminance, (**c**) random $90°$ flip, (**d**) Gaussian noise, (**e**) motion blur, (**f**) random erasure, and (**g**) Mosaic.

Targets in the images were manually labeled using the open-source annotation tool LabelImg [45]. When labeling, the rectangular box fits the citrus contour. For targets occluded by leaves, branches, and other fruits, the approximate outline of the target was labeled according to manual experience. Meanwhile, when the citrus surface area is occluded after more than 80% or the citrus is in the distant background, in principle, the target fruit is not labeled, and the label of the target fruit is named "citrus." The labeled data were saved, XML tag files corresponding to images were generated, and Python scripts were written to convert the XML tag files into txt tag files that the YOLO model could recognize. The annotation is shown in Figure 5.

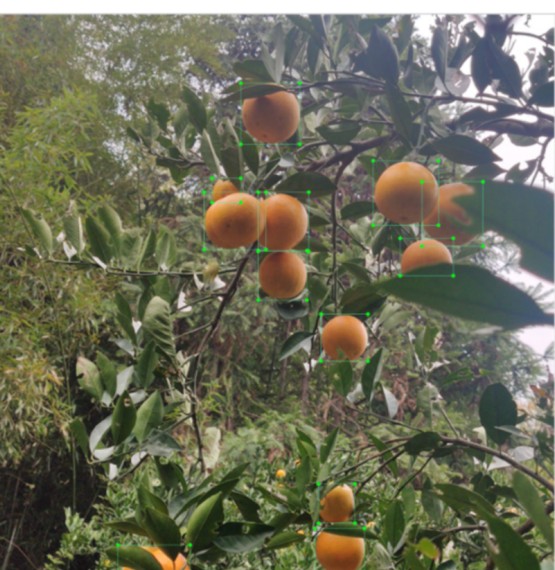

**Figure 5.** Annotate citrus images with LabelImg. Green rectangular box is labeled target box.

### 2.3. Citrus Detection Model YOLO-DCA

2.3.1. Network Architecture

According to the different network depths and widths, YOLOv7 can be divided into seven versions: YOLOv7-tiny, YOLOv7, YOLOv7-X, YOLOv7-W6, YOLOv7-E6, YOLOv7-D6, and YOLOv7E6E. To deploy this model on embedded devices, YOLOv7-tiny was selected as the baseline model in this paper based on the lightweight consideration and the need for accuracy.

YOLOv7 is a single-stage target detection algorithm based on ANCHOR, according to which the YOLOv7-tiny algorithm is streamlined from YOLOv7, retains the composite model scaling method, and uses efficient layer aggregation networks (ELAN) [46] instead of extended efficient layer aggregation networks (E-ELAN), which guarantees the accuracy with a faster number of model parameters and detection speed. This makes it very suitable for the real-time demand of citrus detection and easy to deploy for embedded devices; therefore, in this paper, we choose to improve on YOLOv7-tiny. YOLOv7-tiny consists of three parts: backbone, neck, and head, as shown in Figure 6.

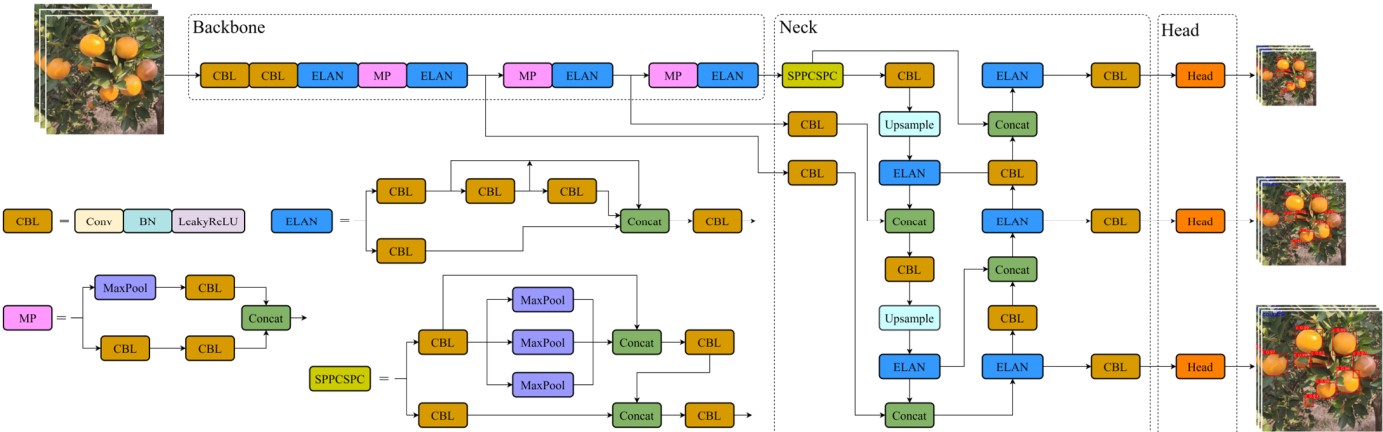

**Figure 6.** YOLOv7-tiny model structure.

Backbone: This mainly consists of CBL, ELAN, MP, and SPPCSPC modules. ELAN mainly consists of VoVNet [47] and CSPNet [48], which solves the problem that the model is difficult to converge during training after scaling and optimizes the gradient length of the whole network. Using SPPCSPC, which is an improvement from SPP (spatial pyramid and pooling) [49], the model achieves feature map fusion of local and global features. It enriches the expressive ability of feature maps and aggregates the feature gradient information, resulting in a reduction of parameters while maintaining faster performance.

Neck: This uses the same path-aggregated feature pyramid (PAFPN) architecture as YOLOv5, combines FPN [50] with PANet [51], a top-down fusion of low-resolution high-semantic information feature maps and high-resolution low-semantic information feature maps, and, at the same time, obtains localization information along the top-down path enhancement.

Head: This uses the feature maps of different scales generated by the Neck part, after three ordinary convolutions, three branches are output to output feature maps of $80 \times 80$, $40 \times 40$, and $20 \times 20$ sizes to detect large, medium, and small targets, respectively, and obtain the final prediction results.

Although YOLOv7-tiny has been streamlined compared to YOLOv7, it still needs many parameters and a complex model. To facilitate the deployment of the model on robot-embedded devices, we propose several modifications to lighten the YOLOv7-tiny network despite its streamlining compared to YOLOv7. Specifically, we replaced the ELAN module in the original YOLOv7-tiny with a lightweight ELAN-DW module, combined the lightweight attention mechanism CA with the CBL module to replace the original CBL module, and replaced the original detector head with the Dynamic Head, which is adapted

to detect targets of different scales by accommodating changes in scale features. The resulting improved YOLOv7-tiny model, named YOLO-DCA, has an enhanced network structure diagram as shown in Figure 7.

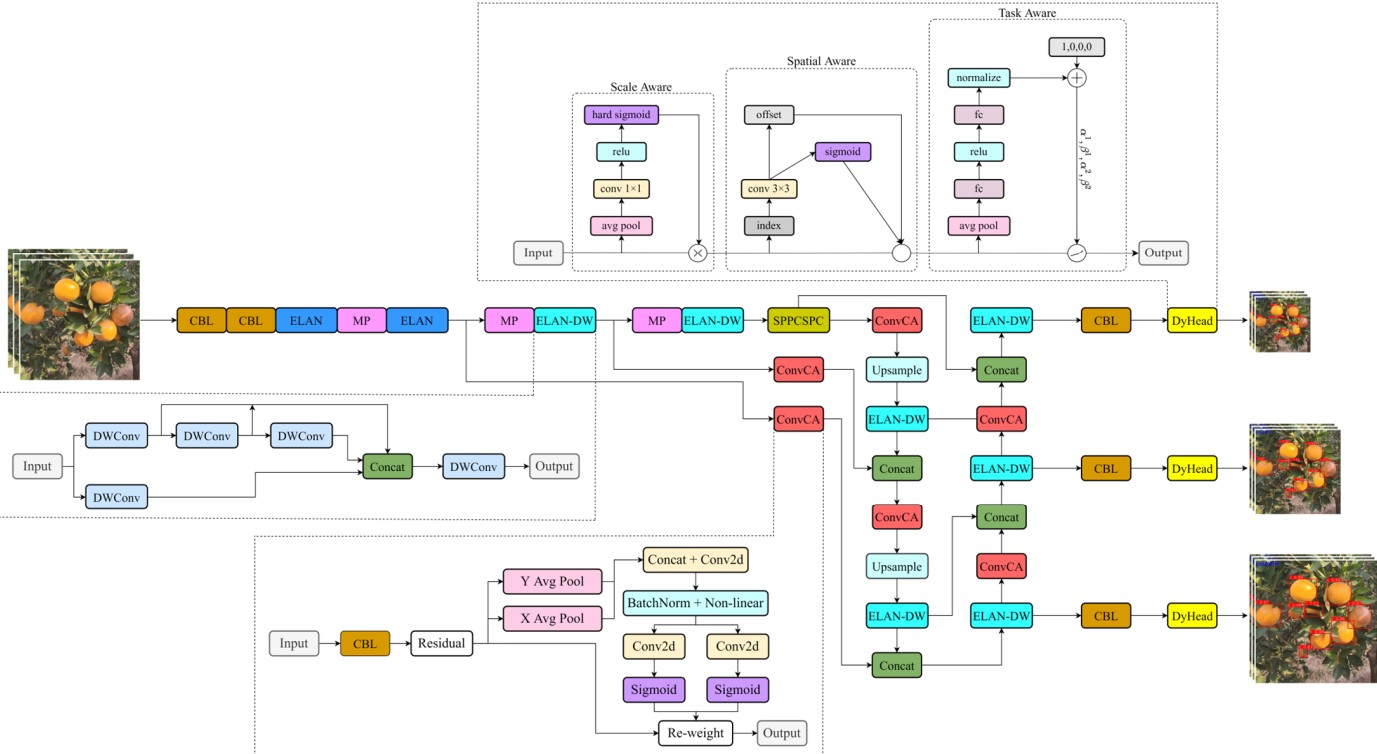

**Figure 7.** YOLO-DCA model structure.

### 2.3.2. Lightweight ELAN Module

ELAN can aggregate multiple features better due to its excellent feature aggregation structure. However, even after YOLOv7-tiny has simplified the original E-ELAN, there is still room for simplification. In order to satisfy the purpose of lightweight, this study uses Depthwise Separable Convolution (DWConv) to replace the ordinary convolution in ELAN, which first appeared in MobileNetV1 and reduces the number of parameters in convolution calculation by splitting the spatial dimension and channel dimension while maintaining the accuracy of the model.

DWConv first uses channel-by-channel convolution (Depthwise Convolution) to split the input N × H × W feature map into N 1 × H × W feature maps, and a convolution kernel is only responsible for calculating the feature maps of one channel, which produces feature maps with the same number of channels as the input channels. Afterward, point-by-point convolution (Pointwise Convolution) is used, which operates in the same way as ordinary convolution, with a convolution kernel size of 1 × 1 × N, with N being the number of channels in the previous layer, after which a weighted summation is performed in the channel dimensions to generate a new feature map, as shown in Figure 8. The replaced ELAN module is shown in Figure 9.

### 2.3.3. CAConv

In deep learning, the more popular attention mechanisms are SE (squeeze-and-excitation network) [52], CBAM (convolutional block attention module) [53], and BAM (bottleneck attention module) [54]. SE only considers the information between encoding channels, ignoring the importance of positional information to the visual task; CBAM and CBM, although they enhance the capture of positional information, cannot effectively capture the information of distant dependencies of the feature map because convolution

only focuses on local information. Coordinate attention (CA) [55], a coordinate attention mechanism, captures spatial information while preserving positional information. Unlike channel attention, which converts the feature tensor into individual feature vectors through 2D global pooling, coordinate attention decomposes the channel attention into two one-dimensional feature encoding processes, which aggregate features along two spatial directions, respectively.

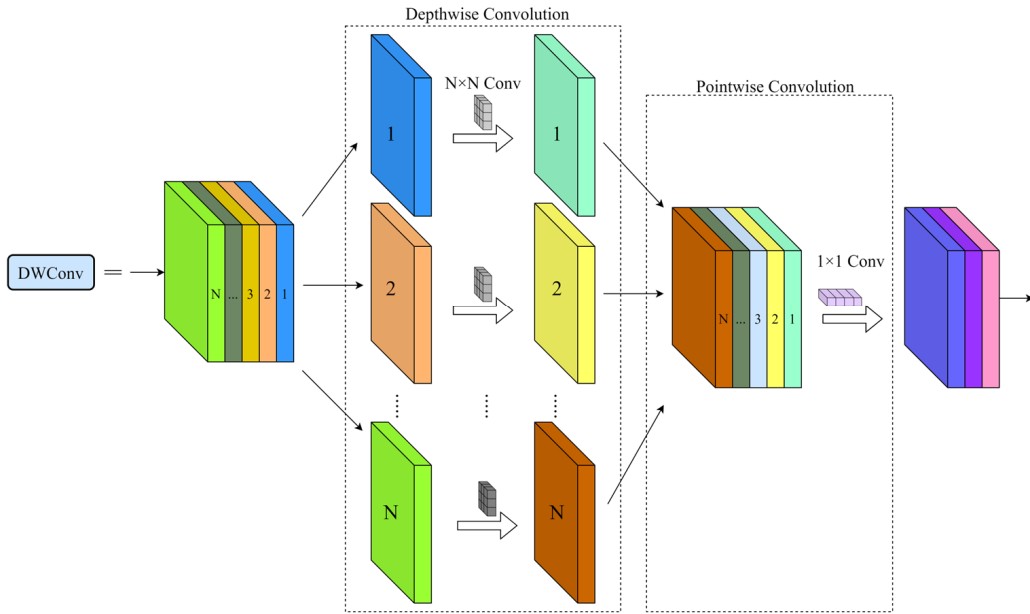

**Figure 8.** DWConv structure.

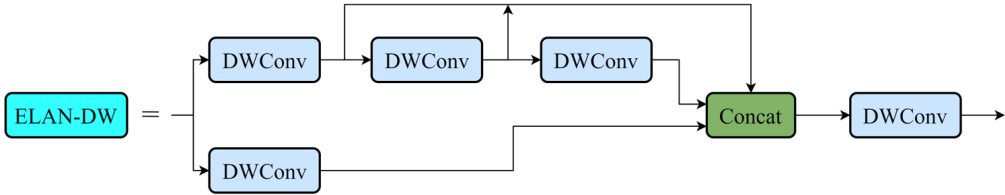

**Figure 9.** ELAN-DW structure.

CA attention consists of two parts: coordinate information embedding and coordinate attention generation. The coordinate information embedding phase encodes each channel along the horizontal and vertical coordinates for a given input X using pooled kernels of (H, 1) and (1, W), respectively, after which features are aggregated along the spatial directions to produce a pair of direction-aware feature maps. The coordinate attention generation phase uses the two feature maps generated by coordinate embedding, then splice, and uses the $1 \times 1$ convolution to compress the channel dimensions. The horizontal and vertical direction information is then encoded using BatchNorm and ReLU activation, after which, along the spatial dimension, the feature maps split again, and the $1 \times 1$ convolution is used to generate the feature maps with the same number of channels as the input X. Finally, they are normalized, weighted, and fused by Sigmoid. Afterward, the output is combined as input to the ordinary convolution to complete the injection of coordinate attention, making it a coordinate attention convolution (CAConv). The CAConv is shown in Figure 10.

### 2.3.4. Dynamic Detection Head

The complexity of classification and localization in target detection tasks requires that the target detection head adapts to features at different scales to detect targets at different scales. Dynamic Head, proposed by Dai et al. [56], enables the target detection

head to be scale-aware, spatial-aware, and task-aware by combining the attention mechanisms between feature hierarchies, spatial locations, and output channels to improve the model performance.

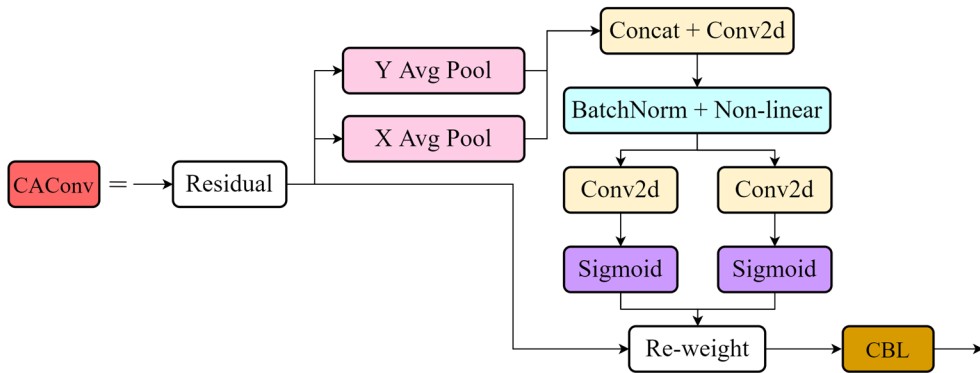

**Figure 10.** CAConv structure.

Before using Dynamic Head, it is necessary to use Feature Pyramid (FPN) to adjust the output features to the same scale to form a 3D tensor $F \in R^{L \times S \times C}$, which uses them as the input of Dynamic Head, and afterward stacks several DyHead blocks sequentially. Among them, the DyHead module consists of three parts: the scale-aware module, the spatial-aware module, and the task-aware attention module. The structure is shown in Figure 11.

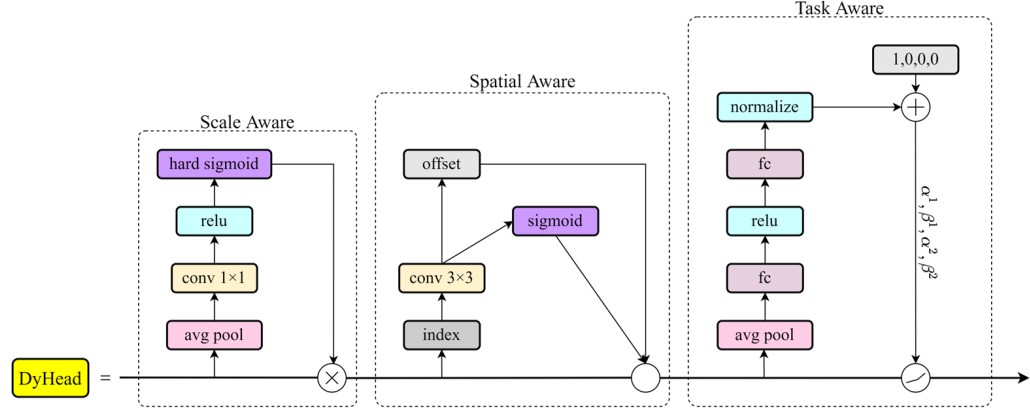

**Figure 11.** DyHead structure.

The scale-aware attention module dynamically fuses features at different scales according to semantic importance. The spatial-aware attention module uses deformable convolution to sparsify the attention, followed by cross-level feature fusion at different locations in the same space for regions with similar target features. The task-aware attention module reduces feature dimensionality through average pooling, followed by dynamic switching of ON and OFF feature channels using two fully connected layers and a normalization function to achieve joint learning and generalization of objects and, finally, task-aware attention.

### 2.4. Experimental Platform and Parameters

We performed all experiments on a platform with the following specifications: CPU: Intel(R) Xeon(R) Platinum 8358P, RAM: 30 GB, GPU: RTX A5000 with 24 GB memory, the Ubuntu version 20.04 operating system, CUDA version 11.3, Python version 3.8, and PyTorch version 1.10. Furthermore, we conducted subsequent comparative experiments on the same platform. For repeating this experiment, this paper recommends using a TITAN Xp and the above graphics card with 6 GB of graphics memory, CUDA version 11.0 and above, and a PyTorch version no less than 1.70.

We set the initial learning rate to 0.01 during training and adjusted the learning rate using cosine annealing decay. We implemented the learning rate preheating strategy for the first three epochs. We utilized the SGD optimizer with a momentum parameter of 0.937 and weight decay of 0.0005 to optimize the network. We trained the model for 300 epochs using a batch size of 64 for the dataset.

*2.5. Evaluation Metrics*

In order to comprehensively evaluate the performance of the model in this paper, precision (P), recall (R), mean average precision (mAP), number of network parameters (Params), model size (MB), gig floating point operations per second (GFLOPs), and detection speed (FPS) are used as evaluation metrics in this paper. P, R, FPS, and mAP are defined as follows:

$$P = \frac{T_p}{T_p + F_p} \times 100\% \tag{1}$$

$$R = \frac{T_p}{T_p + F_n} \times 100\% \tag{2}$$

$$FPS = \frac{1}{T} \tag{3}$$

$$AP = \int_0^1 P(R)dR \times 100\% \tag{4}$$

$$mAP = \frac{\sum AP}{N} \times 100\% \tag{5}$$

where $T_p$ is the number of citrus fruits correctly predicted by the model, $F_p$ is the number of citrus fruits incorrectly predicted by the model, and $F_n$ is the number of citrus fruits omitted to be predicted by the model. P is the proportion of citrus fruits correctly predicted by the model, and R is the proportion of citrus fruits correctly predicted by the model to the total number of citrus fruits. T is the time required by the model to detect a single image. n is the number of detection categories. Since there is only one citrus in this paper, N = 1. AP is the area under the P and R curves. mAP is the average AP value of all citrus categories in the dataset, and in this paper, AP is equal to mAP. Model weights are the magnitude of the weights after the model is trained. Model parameter refers to the number of parameters of the model. GFLOPs refer to gigaflops floating point operations per second, which are used to evaluate the computational complexity of the network. Detection speed refers to the number of images detected by the model per second and is used to assess the real-time detection performance of the model. The accuracy mentioned in this paper refers to mAP0.5.

## 3. Results and Analysis

*3.1. Ablation Experiments*

In order to verify the improvement of the model by each module, we conducted experiments on the YOLOv7-tiny network. We replaced the convolution of the backbone network with DWConv, inserted CA in the neck network, and replaced the original detection head with Dynamic Head. Subsequently, we analyzed and evaluated the experimental results presented in Table 2. Part of the ablation experimental metrics curves are shown in Figure 12.

According to the experimental results, by replacing the ordinary convolution with DWConv, the number of model parameters is reduced by 91.01%, and mAP is reduced by 1.95% compared to the original model. When adding CA to YOLOv7-tiny, the parameters increased by 0.03 M and mAP improved by 1.78%. Similarly, when adding the Dynamic Head in YOLOv7-tiny instead of the original detection head, the parameters increased by 0.04 M and mAP increased by 1.92%. When adding CA to DWConv, mAP increased by

1.85%, and the parameter count decreased by 3.77 M. After using CA and Dynamic Head, mAP increased by 2.41%, and the number of parameters increased by 0.07 M. When using Dynamic Head after DWConv, mAP increased by 0.61%. The parameter count decreased by 3.90 M. When using all three structures simultaneously, mAP improved by 2.09%, and the parameter count decreased by 3.73 M, providing an even more significant overall performance improvement over the original YOLOv7-tiny.

**Table 2.** YOLO-DCA ablation experiment. Where × means this part is not used, √ means this part is used.

| DWConv | CAConv | Dynamic Head | mAP0.5 (%) | Params (M) | GFLOPs |
|--------|--------|--------------|------------|------------|--------|
| × | × | × | 94.79 | 6.01 | 13.2 |
| √ | × | × | 92.84 | 0.54 | 1.5 |
| × | √ | × | 96.57 | 6.04 | 13.2 |
| × | × | √ | 96.71 | 6.05 | 12.5 |
| √ | × | √ | 95.40 | 2.11 | 4.3 |
| × | √ | √ | 97.20 | 6.08 | 12.6 |
| √ | √ | × | 96.64 | 2.24 | 8.0 |
| √ | √ | √ | 96.98 | 2.28 | 6.7 |

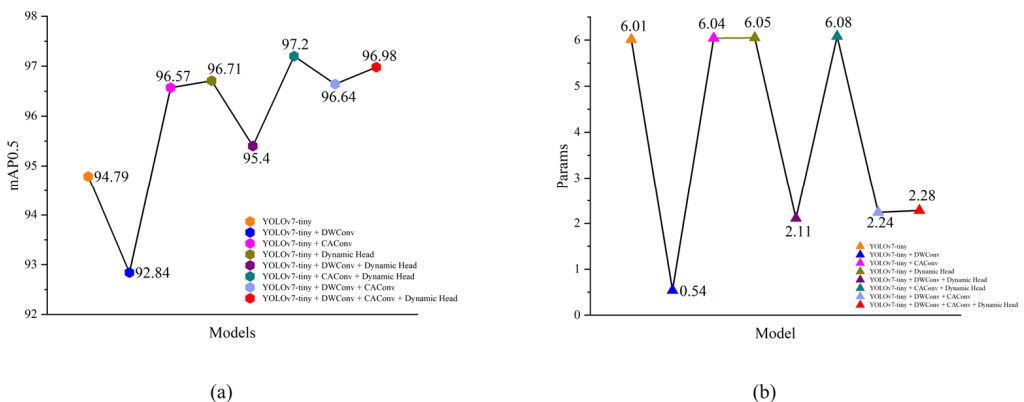

**Figure 12.** Ablation experiment module: (**a**) mAP0.5 curve and (**b**) parameters curve.

### 3.2. Comparison of Lightweight BackBone Networks

In order to compare the effectiveness of the lightweight of this model, a commonly used lightweight network is selected as the backbone network of YOLOv7-tiny, and comparative experiments are conducted in this paper. The experimental results are shown in Table 3. Part of the lightweight backbone experimental metrics curve is shown in Figure 13.

**Table 3.** Comparison of lightweight networks.

| BackBone | P (%) | R (%) | mAP0.5 (%) | Params (M) | GFLOPs |
|----------|-------|-------|------------|------------|--------|
| MobileNetv2 | 93.05 | 88.16 | 95.66 | 4.7 | 10.3 |
| MobileNetv3 | 92.49 | 88.06 | 94.44 | 3.43 | 8.8 |
| GhostNet | 91.20 | 87.97 | 95.08 | 4.23 | 8.6 |
| ShuffleNetv2 | 93.55 | 89.76 | 96.65 | 5.51 | 10.6 |
| Original | 91.05 | 90.55 | 95.02 | 6.01 | 13.2 |
| Improve | 94.12 | 91.55 | 96.38 | 1.53 | 4.0 |

According to the experimental results, when using MobileNetv2, MobileNetv3, Ghost-Net, ShuffleNetv2, and the improved backbone network in this paper as the backbone network of YOLOv7-tiny, the number of parameters and the number of floating-point operations (GFLOPs) are reduced to different degrees compared with the original YOLOv7-tiny. Among them, the backbone network that was improved with the DW-CA structure has

the most significant reduction, with a 74.5% reduction in parameter quantity and a 69.7% reduction in GFLOPs compared to the original model. The proposed network outperforms the other compared models regarding precision and recall and is only 0.27% lower than the best ShuffleNetv2 on mAP0.5. It can be attributed to the CA module's ability to fuse features across locations and channels. However, when other parametric metrics are considered together, this slight decrease in average accuracy has an almost negligible effect on YOLOv7-DCA.

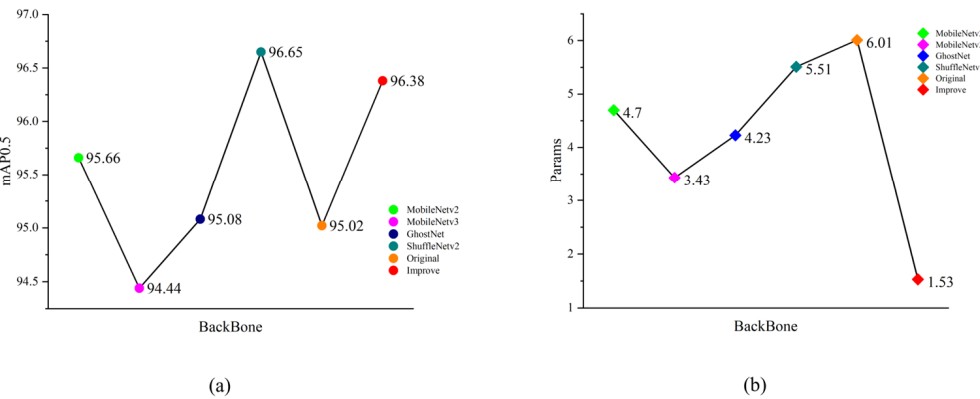

(a)                                                                                                (b)

**Figure 13.** Comparative experimental curves for different lightweight backbone networks: (**a**) mAP0.5 curve and (**b**) parameters curve.

### 3.3. Attention Detection Head Comparison

Embedding an attention mechanism into the detection head part of a target detection model is a commonly used technique that can help the model's detection effectiveness. In this paper, we select the commonly used attention mechanisms, SE, CBAM, and ECA, and insert them into the detection head to verify the effectiveness of Dynamic Head (Dynamic Head) in this model; the experimental results are shown in Table 4. The attention module mAP0.5 metric curve is shown in Figure 14.

**Table 4.** Attention detection head comparison.

| Model | Attetion Module | P (%) | R (%) | mAP0.5 (%) | Params (M) | GFLOPs |
|---|---|---|---|---|---|---|
| YOLOv7-tiny | SE | 93.75 | 91.02 | 96.53 | 6.02 | 13.3 |
| | CBAM | 93.08 | 91.09 | 96.39 | 6.01 | 13.2 |
| | ECA | 92.53 | 90.46 | 96.32 | 6.01 | 13.2 |
| | Dynamic Head | 93.86 | 90.56 | 96.71 | 6.05 | 12.5 |

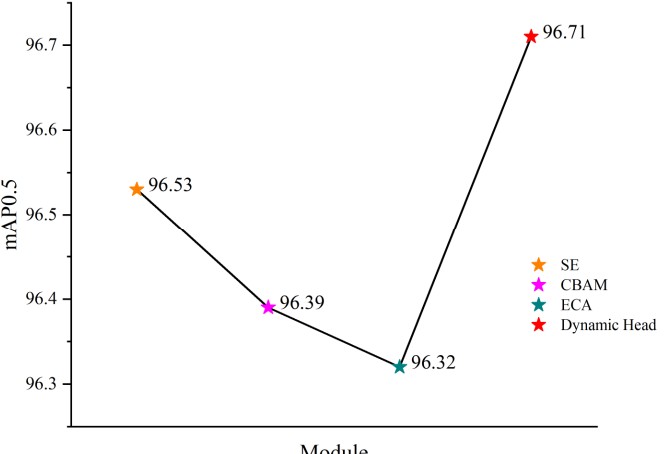

**Figure 14.** Experimental curves for different attention detection heads.

Table 4 shows that the Dynamic Head performs better than other attention detection heads, indicating a more vital generalization ability, robustness, and detection effect. At the same time, the number of parameters added by Dynamic Head is relatively tiny compared to other attention mechanisms, which has little impact on the complexity of the model.

### 3.4. Comparison of Occluded Object Recognition

In citrus orchards in natural environments, it is often the case that fruits overlap each other, and leaves, branches, and weeds obscure fruits. Losing the contour information of the fruit parts increases the difficulty of fruit detection. When the occlusion becomes severe, the fruit contour information is more lost, and the size information of the occluded part after the feature pyramid transfer becomes less and less, making it difficult to detect. Therefore, the model's accuracy in detecting citrus with different levels of occlusion must be analyzed. We selected slightly occluded test set A and severely occluded test set B as the test experimental datasets, and the detection results of the Faster R-CNN, YOLOv3, YOLOv4, YOLOv5s, YOLOX-s, YOLOv6s, and YOLOv7 models are shown in Table 5.

**Table 5.** Comparison of occlusion experiments with different models.

| Model | Test Set | P (%) | R (%) | mAP (%) |
|---|---|---|---|---|
| Faster R-CNN | A | 98.22 | 97.97 | 97.02 |
| | B | 70.21 | 70.11 | 70.48 |
| YOLOv3 | A | 96.88 | 97.97 | 98.30 |
| | B | 64.64 | 64.00 | 84.80 |
| YOLOv4 | A | 99.50 | 99.92 | 99.50 |
| | B | 93.30 | 86.10 | 86.60 |
| YOLOX-s | A | 98.54 | 99.54 | 98.10 |
| | B | 78.78 | 78.52 | 84.20 |
| YOLOv5s | A | 97.90 | 99.60 | 99.50 |
| | B | 94.80 | 83.00 | 90.30 |
| YOLOv6s | A | 99.10 | 98.70 | 99.50 |
| | B | 91.40 | 78.10 | 88.40 |
| YOLOv7-tiny | A | 99.60 | 98.70 | 99.20 |
| | B | 91.40 | 80.60 | 80.10 |
| YOLOv7 | A | 99.60 | 99.93 | 99.90 |
| | B | 97.20 | 87.20 | 87.50 |
| YOLO-DCA | A | 99.60 | 99.95 | 99.95 |
| | B | 94.70 | 88.70 | 89.20 |

The experimental results show that YOLO-DCA performs the best among all the compared models in the slight occlusion scenario. Under a severe occlusion scenario, YOLOv7 has the highest *p* value; it is 2.5% higher than YOLO-DCA. However, under the R and mAP metrics, YOLO-DCA performs best at 88.7% and 89.2%, which is 1.5% and 1.7% higher than YOLOv7, respectively. In summary, compared to other models, YOLO-DCA performs better in the comprehensive metrics.

Table 5 displays the detection results of the three models on the test sets under different occlusion degrees. As shown in Figure 15, slight occlusion does not significantly affect citrus contour and color features, and all models successfully detected the fruit with no leaks. However, Faster R-CNN experienced issues with repeated detection. As shown in Figure 16, with the occlusion increased, all models except YOLO-DCA struggled with recognition difficulties, resulting in leakage, false detection, and repeated detection. Therefore, YOLO-DCA exhibited superior recognition performance for detecting citrus fruit in severe occlusion. The statistical results of some detection with different occlusion degrees are shown in Table 6.

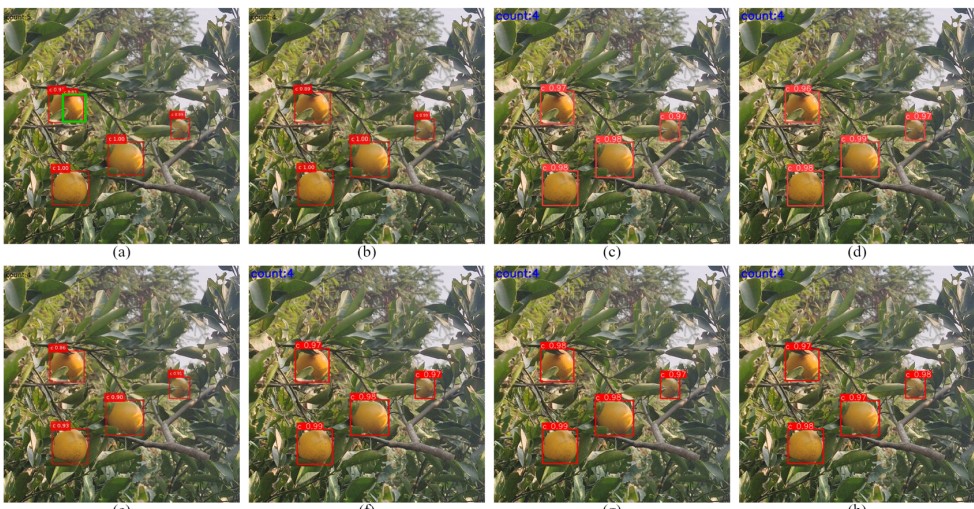

**Figure 15.** Example of different models' detection with slight occlusion. (**a**) Faster R-CNN; (**b**) YOLOv3; (**c**) YOLOv4; (**d**) YOLOv5s; (**e**) YOLOX-s; (**f**) YOLOv7-tiny; (**g**) YOLOv7; (**h**) YOLO-DCA. We used the green rectangular box to mark duplicate detection errors.

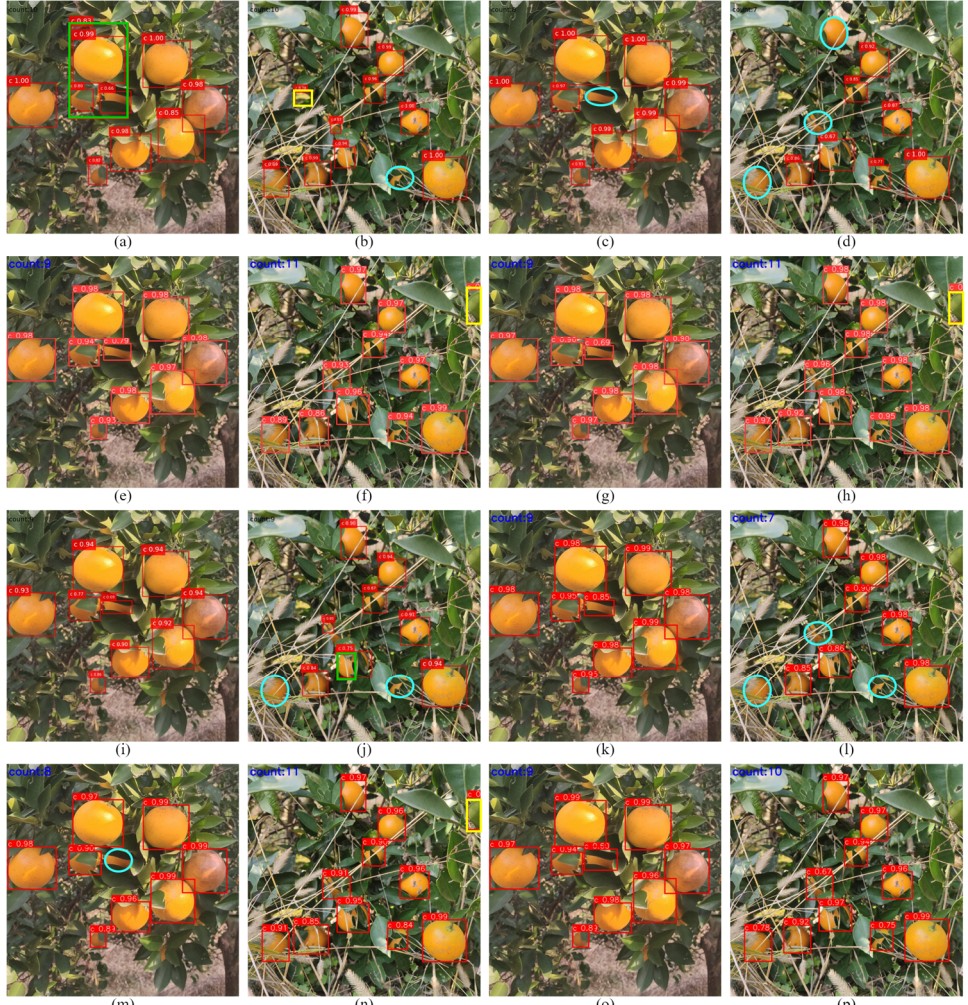

**Figure 16.** Example of different models' detection with several occluded. (**a,b**) Faster R-CNN; (**c,d**) YOLOv3; (**e,f**) YOLOv4; (**g,h**) YOLOv5s; (**i,j**) YOLOX-s; (**k,l**) YOLOv7-tiny; (**m,n**) YOLOv7; (**o,p**) YOLO-DCA. We used the green rectangular box to mark duplicate detection errors, the cyan oval box to mark missed detection errors, and the yellow rectangular box to mark false detection errors.

**Table 6.** Part of different occlusion degree detection number statistics.

| Model | Degree of Occlusion | Real Nums. | Detect Nums. | Mean Confidence Score |
|---|---|---|---|---|
| Faster R-CNN | | 4 | 5 | 0.95 |
| YOLOv3 | | 4 | 4 | 0.97 |
| YOLOv4 | | 4 | 4 | 0.97 |
| YOLOv5s | Slightly | 4 | 4 | 0.97 |
| YOLOX-s | | 4 | 4 | 0.90 |
| YOLOv7-tiny | | 4 | 4 | 0.97 |
| YOLOv7 | | 4 | 4 | 0.98 |
| YOLO-DCA | | 4 | 4 | 0.97 |
| Faster R-CNN | | 19 | 20 | 0.88 |
| YOLOv3 | | 19 | 15 | 0.72 |
| YOLOv4 | | 19 | 20 | 0.89 |
| YOLOv5s | Several | 19 | 20 | 0.90 |
| YOLOX-s | | 19 | 20 | 0.90 |
| YOLOv7-tiny | | 19 | 16 | 0.79 |
| YOLOv7 | | 19 | 18 | 0.89 |
| YOLO-DCA | | 19 | 19 | 0.91 |

*3.5. Comparison of Detection under Different Light Conditions*

To test the robustness of models in different illumination angles, we selected 30 front-light, 30 side-light, and 30 dark-light images from the mildly occluded dataset A, severely blocked dataset B, and the validation set. We then used them as experimental materials. Table 7 and Figure 17 show the corresponding test results. YOLO-DCA's comprehensive performance in front-light and side-light scenarios is slightly lower than that of YOLOv7. Still, both of them are better than YOLOv7 in dark–light environments. However, the combined performance of all three models decreases in dark–light situations, affecting the model's detection accuracy due to the loss of fruit color and texture features caused by the lack of light.

**Table 7.** Comparison of different illumination angle detection.

| Illumination Angle | Model | P (%) | R (%) | mAP (%) |
|---|---|---|---|---|
| | YOLOv7-tiny | 94.50 | 86.70 | 86.70 |
| Front-light | YOLOv7 | 96.80 | 90.10 | 90.80 |
| | YOLO-DCA | 96.10 | 88.90 | 89.10 |
| | YOLOv7-tiny | 95.40 | 87.70 | 88.40 |
| Side-light | YOLOv7 | 96.80 | 88.90 | 89.30 |
| | YOLO-DCA | 96.20 | 90.10 | 89.80 |
| | YOLOv7-tiny | 91.10 | 78.90 | 78.50 |
| Dark-light | YOLOv7 | 94.10 | 80.20 | 80.30 |
| | YOLO-DCA | 93.70 | 80.80 | 80.60 |

Nevertheless, the combined performance of YOLO-DCA is the highest among all models. It indicates that YOLO-DCA is more adaptable to scenes with complex lighting and robust to changes in illumination angles. Figure 17 shows that YOLOv7-tiny has a leakage problem in the front-light background, while citrus presents a clear texture in the Side-light scene, and there is no leakage of signatures in all models. YOLOv7 has a leakage problem in the Dark-light background. However, YOLO-DCA had no leakage problems in the above three scenarios.

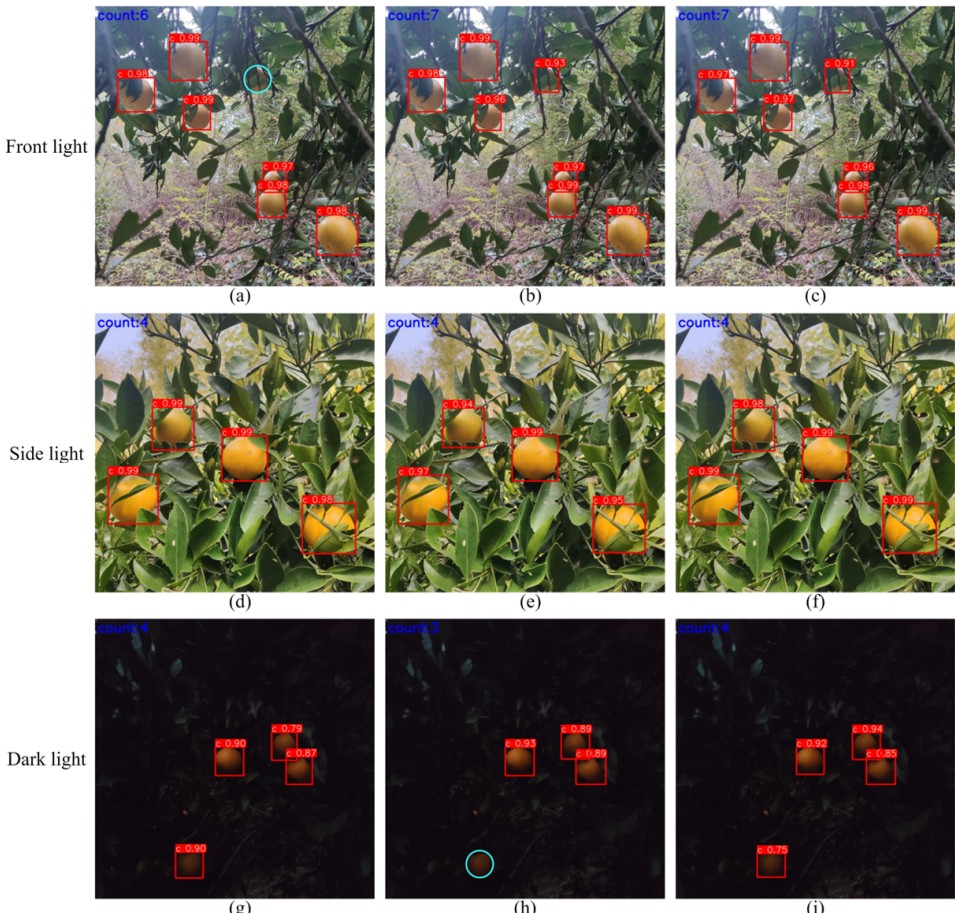

**Figure 17.** Detection results of different light conditions. Front-light: (**a**) YOLOv7-tiny; (**b**) YOLOv7; (**c**) YOLO-DCA. Side-light: (**d**) YOLOv7-tiny; (**e**) YOLOv7; (**f**) YOLO-DCA. Dark-light: (**g**) YOLOv7-tiny; (**h**) YOLOv7; (**i**) YOLO-DCA. We used the cyan oval box to mark missed detection errors.

### 3.6. Comparison of Different Blur Detection

To verify the fault tolerance of YOLO-DCA for blurred images, we randomly selected 30 shots from the unenhanced mildly occluded dataset A and the severely occluded dataset B, totaling 60 images. These images simulate problems in outdoor working environments, such as dirty lenses, mechanical shake, and ambient light variations. In this paper, we investigate the lens shake at different motion movement speeds of the picking robot by applying motion blur operation and setting the blur radius to 11, 31, and 51. Table 8 displays the statistical results; Figure 18 displays the test results.

**Table 8.** Comparison of different blur detection.

| Radius of Blur | Model | P (%) | R (%) | mAP (%) |
|---|---|---|---|---|
| | YOLOv7-tiny | 93.90 | 75.00 | 74.60 |
| 11 | YOLOv7 | 97.00 | 78.20 | 78.30 |
| | YOLO-DCA | 97.00 | 79.00 | 79.20 |
| | YOLOv7-tiny | 94.90 | 60.50 | 59.90 |
| 31 | YOLOv7 | 93.60 | 58.90 | 57.90 |
| | YOLO-DCA | 95.00 | 61.30 | 60.90 |
| | YOLOv7-tiny | 76.50 | 41.90 | 38.70 |
| 51 | YOLOv7 | 79.10 | 42.70 | 39.60 |
| | YOLO-DCA | 84.40 | 43.50 | 41.40 |

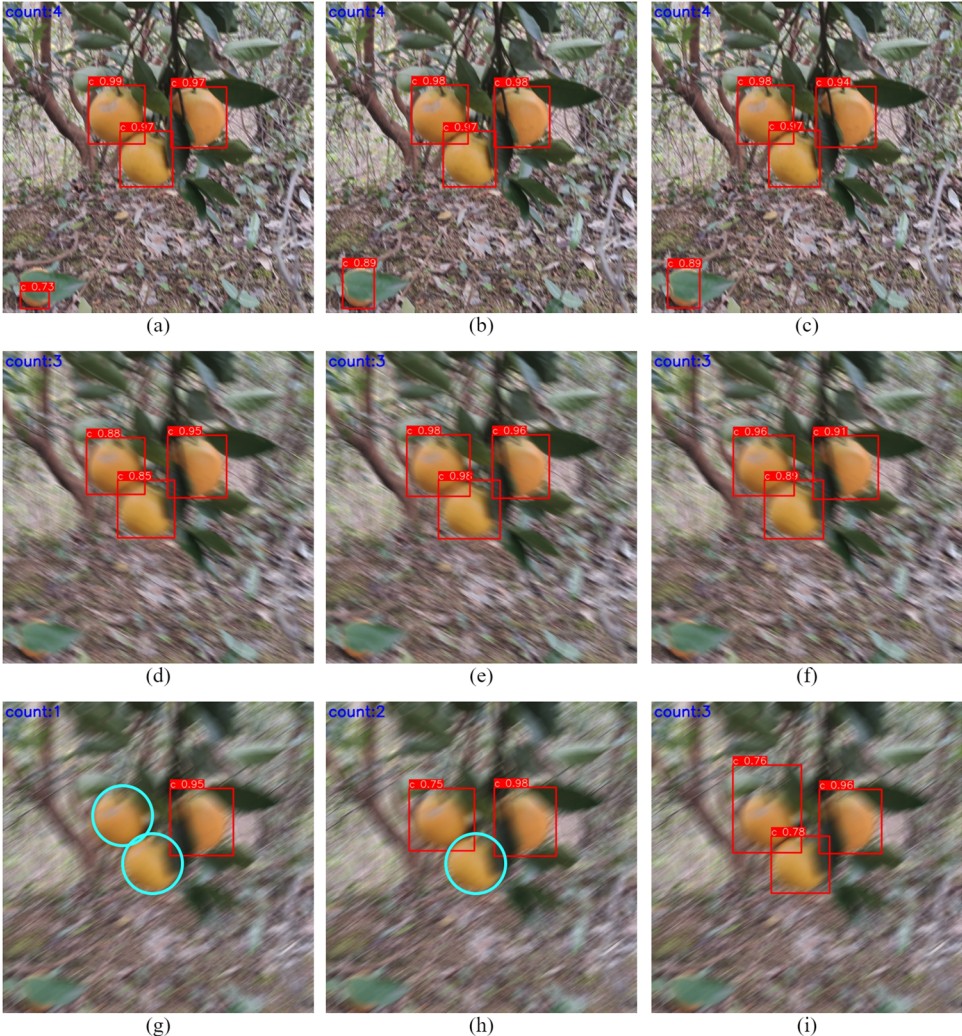

**Figure 18.** Detection results for different blur radii. Radius of 11: (**a**) YOLOv7-tiny; (**b**) YOLOv7; (**c**) YOLO-DCA. Radius of 31: (**d**) YOLOv7-tiny; (**e**) YOLOv7; (**f**) YOLO-DCA. Radius of 51: (**g**) YOLOv7-tiny; (**h**) YOLOv7; (**i**) YOLO-DCA. We used the cyan oval box to mark missed detection errors.

The analysis of the experimental data shows that YOLOv7-tiny, YOLOv7, and YOLO-DCA can accurately detect the target at a blur radius of 11, and YOLOv7 and YOLO-DCA can also accurately identify some targets at a blur radius of 31. However, in the case of a blur radius of 51, YOLOv7-tiny and YOLOv7 cannot remember the main target correctly, while YOLO-DCA can still identify the target accurately. These results show that YOLO-DCA exhibits strong fault tolerance in the face of blurred images.

### 3.7. Visual Analysis of Model

Convolutional neural networks can only obtain detection results when dealing with target detection problems but could not be more interpretable in network processing. Therefore, this study uses visual activation heatmaps to compare the visualization of YOLOv7-tiny, YOLOv7, and the improved YOLO-DCA model to visualize the features extracted after the last convolution for citrus detection. The darker the red region in the heatmap, the more significant the impact of the location on detection and differentiation. As shown in Figure 19, the feature extraction ability of YOLO-DCA is generally more robust than that of the unimproved YOLOv7-tiny as well as YOLOv7 under different lighting conditions and different disturbing factors. The proposed lightweight citrus target

detection model YOLO-DCA is more suitable to be deployed to the citrus picking robotic terminal for citrus recognition.

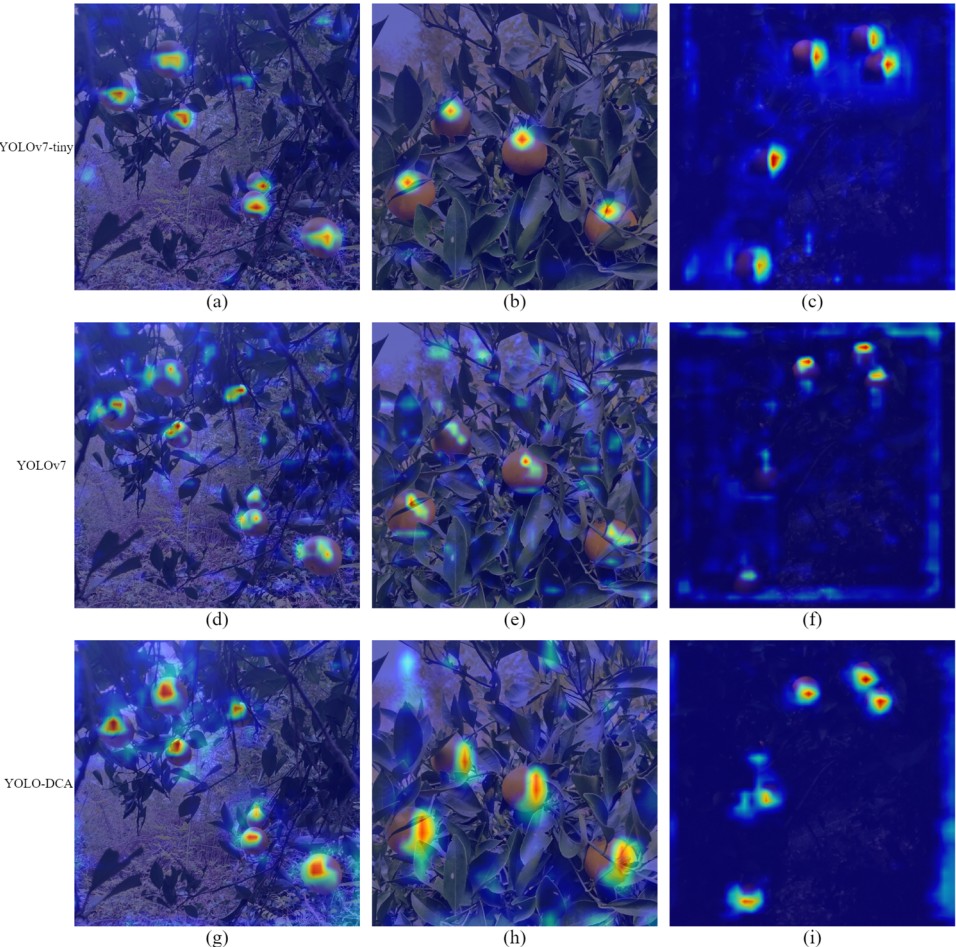

**Figure 19.** Results of different model visualizations: (**a**) moreover, (**b**,**c**) are YOLOv7-tiny; (**d**) moreover, (**e**,**f**) are YOLOv7; (**g**) moreover, (**h**,**i**) are YOLO-DCA. The darker the color in the diagram, the more the model focuses on this area.

### 3.8. Comparison of Different Models

To verify the performance of YOLO-DCA, 16 models of Faster R-CNN, YOLOv3, YOLOv4, YOLOv5 series, YOLOv6 series, YOLOX-tiny, YOLOX-S, YOLOv7-tiny, YOLOv7, and YOLO-DCA were selected for comparison. Table 9 shows the results. Part of the metric curve of the comparison models is shown in Figure 20.

According to the experimental results, YOLO-DCA is second only to YOLOv5n in terms of GFLOPs, number of parameters, and model size, with an increase of 2.7 GFLOPs, 0.33 M, and 0.8 MB, respectively. Still, it has a better overall performance with a rise of 1.8% and 75.4 FPS in mAP and detection rate, respectively, than YOLOv5n. In addition, YOLO-DCA is second only to YOLOv7 in terms of mAP, which is only 0.15% lower than YOLOv7, but the GFLOPS and Params of YOLO-DCA are 93.5% and 94.24% lower than YOLOv7. Meanwhile, in terms of detection rate, YOLO-DCA has the fastest detection rate among all compared models, with a detection rate of 169.8 FPS.

This paper also compares recent fruit detection methods using YOLOv7 or YOLOv7-tiny despite the existence of different factors such as datasets, experimental environments, hardware equipment, and hyperparameters, which make it impossible to make a fair comparison between related studies, since this paper uses a similar methodology with other researchers, and the subjects of the study are more similar. The evaluation metrics

used are similar. We present a discussion of their research results. Table 10 compares the relevant performance indicators.

**Table 9.** Comparison of different mainstream models.

| Model | mAP0.5 (%) | FPS (GPU) | GFLOPs | Params (M) | MS (MB) |
|---|---|---|---|---|---|
| Faster R-CNN | 85.40 | 51.60 | 91.03 | 41.10 | 315.00 |
| YOLOv3 | 92.33 | 56.30 | 77.54 | 61.52 | 469.80 |
| YOLOv4 | 94.60 | 73.20 | 52.90 | 24.34 | 245.90 |
| YOLOv5n | 95.18 | 94.40 | 4.20 | 1.77 | 3.70 |
| YOLOv5s | 94.72 | 153.50 | 15.90 | 7.02 | 13.80 |
| YOLOv5m | 94.80 | 108.70 | 47.90 | 20.85 | 40.30 |
| YOLOv5l | 95.10 | 86.60 | 107.60 | 46.11 | 88.60 |
| YOLOv5x | 94.15 | 58.40 | 203.80 | 86.10 | 165.10 |
| YOLOX-tiny | 94.21 | 59.30 | 7.570 | 5.03 | 58.10 |
| YOLOX-s | 91.65 | 51.10 | 13.32 | 8.94 | 102.90 |
| YOLOv6n | 95.70 | 140.10 | 11.34 | 4.63 | 9.90 |
| YOLOv6s | 95.70 | 123.70 | 45.17 | 18.50 | 38.70 |
| YOLOv6m | 95.90 | 84.10 | 85.62 | 34.80 | 72.50 |
| YOLOv6l | 95.90 | 67.90 | 150.40 | 59.54 | 114.10 |
| YOLOv7-tiny | 94.79 | 163.50 | 13.20 | 6.01 | 11.70 |
| YOLOv7 | 97.13 | 101.40 | 103.20 | 36.48 | 71.30 |
| YOLO-DCA | 96.98 | 169.80 | 6.70 | 2.10 | 4.50 |

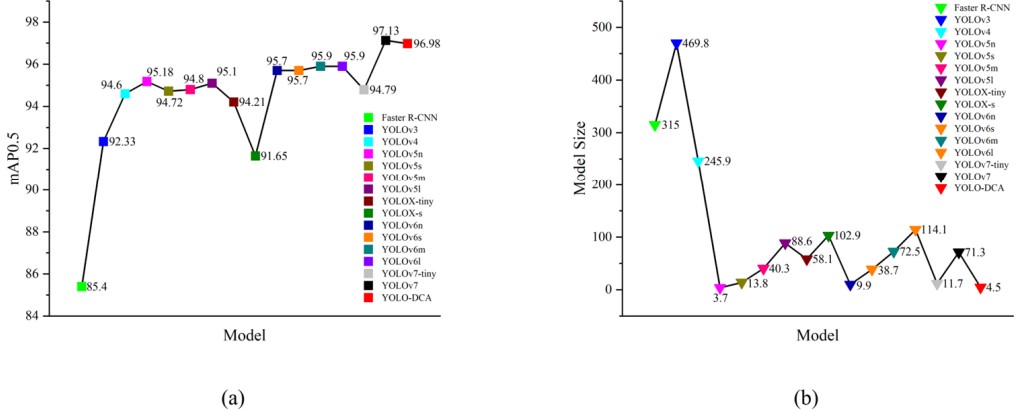

(a)

(b)

**Figure 20.** Comparison curves for different models: (**a**) mAP0.5 and (**b**) model size.

**Table 10.** Comparison results with other works models.

| Authors | Crop | Number of Images | Image Size | Models | mAP0.5 (%) | FPS (GPU) | Params (M) | MS (MB) |
|---|---|---|---|---|---|---|---|---|
| Ma et al. [57] | MinneApple | 841 | 320 × 320 and 410 × 410 | YOLOv7-tiny | 80.40 | 101.01 | / | 5.06 |
| Zhang et al. [58] | Citrus | 1640 | 4864 × 3648 | YOLOv7-tiny | 90.34 | 128.83 | 1.02 | 3.98 |
| Liu et al. [59] | Yellow Peaches | 1021 | 4000 × 3000 | YOLOv7 | 80.40 | 21.00 | / | 51.90 |
| Our | Yongxing Bingtang Citrus | 1908 | 640 × 640 | YOLOv7-tiny | 96.98 | 169.80 | 2.10 | 4.50 |

The table is incomplete as some data are not in the related literature.

Ma et al. [57] used an improved YOLOv7-tiny to detect small apple targets using the public dataset MinneApple, and compared to their model, YOLO-DCA has advantages in mAP, detection speed, number of parameters, and model size. Zhang et al. [58] used a DJI Phantom 4 RTK quadrotor UAV to collect citrus images from Sihui City, Guangdong

Province, at an altitude of 50 m above the ground, and improved YOLOv7-tiny for the study. Compared with their model, YOLO-DCA is 6.64% and 40.97 higher in mAP and detection speed, respectively, while Params and Model size only increased by 1.08 M and 0.52 MB. Liu et al. [59] used yellow peach images collected from the yellow peach plantation in Daping Village, Jinggangshan City, Jiangxi Province, and improved YOLOv7-tiny to conduct the study. Compared with their model, YOLO-DCA has advantages regarding mAP, detection speed, and model size. Compared with the above models, YOLO-DCA balances the detection speed, model parameters, and model size while improving the accuracy.

Combining the above results, YOLO-DCA has better overall performance among all the compared models, ensures the accuracy of the model while significantly reducing the number of model parameters and computation, and has a relatively high detection speed, which makes it more suitable for edge devices, resource-constrained platforms, and the deployment of mobile applications.

## 4. Discussion

Although the model proposed in this paper can achieve accurate and fast detection of citrus, it will still have some limitations. (1) The data type is mature citrus fruits; green and yellow-green citrus were not studied. (2) The accuracy of partial fruit detection in severe occlusion is rather average. (3) No practical experiments were conducted by deploying this model at edge devices. In future research, we will focus on obtaining more pictures of green and yellow-green citrus, combine them with mature citrus to form a dataset, and divide them into separate categories to detect the citrus and evaluate the maturity of the citrus at the same time. Further research will be conducted on multi-source information fusion methods to improve the accuracy in case of severe occlusion and actual deployment of the present model to edge devices for experiments.

## 5. Conclusions

To achieve fast and accurate detection of citrus in complex environments and to deploy the model in edge devices with limited computational resources, this paper proposes an improved citrus detection model, YOLO-DCA. The proposed model reconstructs an efficient layer aggregation network module using depth-separable convolution, integrates ordinary convolution and CA attention, and uses the dynamic detection head. Experimental results show that YOLO-DCA outperforms the original YOLOv7-tiny. The GFLOPs decreased from 13.2 G to 6.7 G, and the Params decreased from 6.01 M to 2.27 M, which is a decrease of 49.24% and 62.23%, respectively. The memory space occupied by the model was also reduced from 11.7 MB to 4.5 MB, which is a decrease of 61.53%. Meanwhile, the mAP of YOLO-DCA improved by 2.19% from 94.79% to 96.98%. This indicates that based on multi-method improvement, YOLO-DCA can detect citrus accurately in complex environments and devices with limited computational resources.

**Author Contributions:** Conceptualization, H.S., C.W., B.G., X.L., Y.H. (Yingjian Hou) and Y.H. (Yuanhui Hu); methodology, B.G.; software, B.G.; validation, B.G. and H.S.; formal analysis, B.G., H.S. and C.W.; writing—original draft preparation, B.G.; writing—review and editing, H.S., C.W., X.L., Y.H. (Yingjian Hou) and Y.H. (Yuanhui Hu); visualization, B.G.; project administration, H.S.; funding acquisition, H.S. and C.W. All authors have read and agreed to the published version of the manuscript.

**Funding:** This research was supported by the National Natural Science Foundation of China–Joint Fund (u19a2061) and Jilin Provincial Development and Reform Commission Capital Construction Funds (Innovation Capacity Building) Project: Research on Key Technologies for Remote Sensing Crop Phenotype Data Analysis (No. 2021C044-8).

**Data Availability Statement:** The datasets in this study are available from the corresponding author on reasonable request.

**Conflicts of Interest:** The authors declare no conflict of interest.

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
