# Peer review of "Improved YOLOv7-Tiny Complex Environment Citrus Detection Based on Lightweighting"

_agronomy, doi:10.3390/agronomy13112667_

Round 1

Reviewer 1 Report

This manuscript entitled " Improved YOLOv7-tiny complex environment citrus detection based on lightweighting" developed a citrus detection model YOLO-DCA. Overall, the authors did a lot of work, and the performance of the proposed algorithm was greatly improved. The detail comments are as follows.
1. In line 16, the fully spelled mAP is not mean accuracy.
3. In lines 65-66, could the author explain why a large amount of memory is conducive to practical deployment?
4. In lines 73-75 and 76-78, the detection speed and model's training samples should be stated. 
5. In table 1, the author discussed the results of 7 combinations for ablation experiment. Could the author explain why the result for original backnone+CAConv+Dynamic Head was not discussed?

1. In line 48, could the author double check the confused sentence "Two-stage detection algorithms Region generation is performed first"? 

2. In line 79, could the author double check the confused sentence "Liu et al. [44], an improved YOLOv5 to identify and count citrus fruits, achieved 98.40% AP". 

3. In line 207, the phrase "model accuracy" as a sentence was confused. 

4. Could the author correct tense to make sure they are consistent throughout the paper? For example, the past tense "increased" in line 319 but the present tense "decreases" in line 322, "selected" in line 443 but "act" in line 447.

5. In line 419, what does Fig. xx represent for? 

6. In lines 502-503, the sentence has no subject.

Author Response

Our response to you is in the attached word document.

-----------

Response to Reviewer 1 Comments

Thank you for your valuable comments, we have made changes and responses according to them. If we still can't answer your question, please let us know and we will be sure to correct it in time. Here is our response:

Point 1: In line 16, the fully spelled mAP is not mean accuracy.

Response 1:

Thank you very much for your valuable comments. Regarding the complete spelling of mAP, it is indeed an oversight on my part; the correct complete spelling of mAP should be "Mean Average Precision", which I have corrected in the original text. Thank you for your correction.

Point 2: In lines 65-66, could the author explain why a large amount of memory is conducive to practical deployment?

Response 2:

Thank you very much for your valuable comments. The phrase "a large amount of memory is conducive to practical deployment" is due to my mistake; the actual situation should be "small memory is conducive to practical deployment." I have revised the corresponding position in the original text. Thank you for your correction.

Point 3: In lines 73-75 and 76-78, the detection speed and model's training samples should be stated.

Response 3:

Thank you very much for your valuable comments. We have added detection speeds to the studies of Huo et al. [42] as well as Xinyang et al. [43] in lines 73-75, as well as describing the source of their model training data.

Huo et al. [42] used " Shantanju " citrus collected from Conghua, Guangzhou, and im-proved the YOLOv5s algorithm to detect and locate mature citrus. The recall rates under uneven, weak and good lighting were 99.55%, 98.47% and 98.48 respectively. The average detection time per frame is 78.96 ms. However, the detection speed of this method needs to be improved.

Xinyang et al. [43] improved YOLOv5s by introducing ShuffleNetV2, SimAM attention and Alpha-IoU using images taken from Foshan citrus orchards in Guangdong Province as well as pictures acquired from the web and proposed the YO-LO-DoC citrus detection method, with a P-value and a mAP value of 98.8% and 99.1%, respectively, and an FPS of 187. However, the model's training samples need to be in-creased.

Point 4: In table 1, the author discussed the results of 7 combinations for ablation experiment. Could the author explain why the result for original backnone+CAConv+Dynamic Head was not discussed?

Response 4:

Thank you very much for your valuable comments. Indeed, we did not add the experimental results of backbone+CAConv+Dynamic Head, which is a deficiency in our work. We said this ablation experiment in Table 2 of the original article and the resultant curves in Figure 12. Thank you for your correction.

Point 5: In line 48, could the author double check the confused sentence "Two-stage detection algorithms Region generation is performed first"?

Response 5:

Thank you very much for your valuable comments. We have scrutinised and identified this problem, and in the original text, we have amended this paragraph to read as follows: "The two-stage detection algorithm first generates candidate regions that may contain the target through several strategies, and these candidate regions are then passed through a convolutional neural network to classify samples to determine if they contain the target object [23].". Thank you for your correction.

Point 6: In line 79, could the author double check the confused sentence "Liu et al. [44], an improved YOLOv5 to identify and count citrus fruits, achieved 98.40% AP".

Response 6:

Thank you very much for your valuable comments. We have scrutinised and identified this problem, and in the original text, we have amended this paragraph to read as follows: “Liu et al. [44] CA attention, replacing PAFPN with BiFPN, and using the zoom loss function, YOLOv5 was improved to detect four varieties of citrus collected from Lingui City, Guilin, Guangxi, namely, “Kumquat”, “Nanfeng tangerine”, “Fertile tangerine”, and “Sha-tang tangerine”. “Tangerine”, “Fertile tangerine”, and “Shatang tangerine” were collected from Lingui City, Guilin, Guangxi Province, and the mAP was 98.4%, 98.4%, and the detection time for a single image is 19ms. However, the number of parameters for this model is significant.”. Thank you for your correction.

Point 7: In line 207, the phrase "model accuracy" as a sentence was confused.

Response 7:

Thank you very much for your valuable comments. We On reviewing the original text, we find that "model accuracy." is redundant, and we have deleted it from the original text. Thank you for your correction.

Point 8: Could the author correct tense to make sure they are consistent throughout the paper? For example, the past tense "increased" in line 319 but the present tense "decreases" in line 322, "selected" in line 443 but "act" in line 447.

Response 8:

Thank you very much for your valuable comments. After reviewing the original text, we found an inconsistency in tenses, and we have now normalised the tenses. For your general present tense "decreases" in line 319, we have changed it to the past tense "decreased"; and for "select" in line 443, we have changed it to "selected".

Point 9: In line 419, what does Fig. xx represent for?

Response 9: Thank you very much for your valuable comments. The Fig. xx in line 419 stands for Figure 17. We have amended the paper to read "Figure 17 correctly". Thank you for your correction.

Point 10: In lines 502-503, the sentence has no subject.

Response 10:

Thank you very much for your valuable comments. After carefully checking lines 502-503, we find that the subject is indeed missing, and we modify the sentence in such a way that the modified sentence reads, “The proposed model reconstructs an efficient layer aggregation network module using depth-separable convolution, integrates ordinary convolution and CA attention and uses the dynamic detection head.”. Thank you for your correction.

Reviewer 2 Report

Comments can be found in the attached pdf file

Author Response

Our response to you is in the attached word document.

-----------

Response to Reviewer 2 Comments

We found someone who is fluent in English to correct our language problem.

Thank you for your valuable comments, we have made changes and responses according to them. If we still can't answer your question, please let us know and we will be sure to correct it in time. Here is our response:

Point 1: A taxonomy domain should be specified regarding which species of citrus authors did cover in their algorithms since it might affect shape, color, etc. Given hundreds of species, if “citrus” refers to any kind of that fruit, the dataset might be too small; if the dataset refers only to few particular species (which I think is the case), then such species should be specified;

Response 1:

Thank you very much for your valuable comments. In the data collection section, we did miss the description of the citrus variety, and the data used in this study only relate to a specific citrus variety, which is the "Yongxing Bingtang Citrus" from Yongxing County, Chenzhou City, Hunan Province, China. In the data collection section, we have added the missing information. Thank you for your correction.

Point 2: Regarding blur tolerance: did authors considered the possibility of “stacking” the images before passing them to the detection algorithm? It is a technique which is common in environment with high blur or ISO noise (i.e. pictures of planets from telescopes) and that reproduces a better image from different (several, sometimes) blurred images. Since the detection rate seems to be very high, spending a couple frames for getting one single image with limited blur can be feasible. What is your point of view? If blur could be detected I also think it would be a valuable source of information (it would mean that algorithm response would not be “no citrus”, but “too blurry to detect”);

Response 2:

Thank you very much for your valuable comments. Regarding your idea of "stacking images", we used the mixup technique in our initial experiments, i.e., we set different transparency settings for multiple training sample images and then stacked them together to form a new training sample image, as shown in Figure 1. After the experiment, we found that its detection performance on the model was more perturbing. The model accuracy appeared to be degraded, as shown in Table 1. As shown in Figure 1, problems such as omission and duplicate detection appeared on the test sample images. After reviewing the literature on fruit detection, especially citrus detection, it was found that the number of researchers using this technique was low. After discussing with the other authors of this paper and asking experts in the field, we decided not to use this technique but to use Gaussian fuzzy as well as motion fuzzy to be more in line with the operational reality.

Figure 1. Part of training samples after mixup augmentation

As shown in Figure 1, this image is the image after we enhanced it using mixup before training. The blue rectangular box is the original labelling information box that we labelled for the presence of citrus targets in the original image. In Figure 1, we can see that when we use mixup, the translucent citrus target will be similar in colour to the dead leaves and weeds in the background, making the model learning tricky and confusing the background and target information, the valuable feature information obtained by the model will be reduced.

Table 1. Experimental results with mixup and without mixup

Models

mAP0.5(%)

YOLOv7-tiny-mixup

92.92

YOLOv7-tiny-no-mixup

94.79

YOLO-DCA-mixup

94.72

YOLO-DCA-no-mixup

96.98

As can be seen from Table xx, the models all showed varying degrees of accuracy degradation after using mixup.

Figure 2. Experimental test results

As shown in Figure 2, after training with mixup, the models all showed varying degrees of leakage. YOLO-DCA without mixup addition, as well as YOLOv7-tiny, showed no leakage; after adding mixup, both models showed leakage, which indicates that the addition of mixup has interfered with the ability of the model to learn features. We hope that our answers have addressed your concerns. Thank you for your correction.

Point 3: Lines 39-40: even weather/environmental conditions could be included to the list, since rain, strong winds or dust can affect the capabilities of the camera to provide good material;

Response 3:

Thank you very much for your valuable comments. In addition, the detection accuracy is low, and the real-time generalization ability is poor in the case of changing lighting conditions, the presence of occlusion of the fruit [10], the similarity of the fruit and the background colour [11-12], and even weather and environ-mental changes.

Regarding the situation of "rain, strong wind or dust", as this study is based on a conventional scenario, we asked agronomy experts and actual fruit farmers when we collected the data. They gave us the opinion that rain, strong wind and dust are not suitable for collection because it will affect the quality of the fruits and thus affect the sales of fruits. Sales. In addition, at the time of our collection, the local weather was mainly cloudy, cloudy and sunny, which was the peak season for citrus collection. It was more difficult for the researcher to collect pictures if rainy days, strong winds and dusty weather conditions were involved, in addition to missing the peak season for collection, which resulted in an insufficient number of samples and poor representativeness. We hope that our answers have addressed your concerns. Thank you for your correction.

Point 4: Line 41: “In addition” is repeated and in general infrared or near-infrared are included in most of the multispectral imaging. It could be omitted or stated like “Costs can also be an issue if citrus detection is performed by hyperspectral or multispectral image analysis”;

Response 4:

Thank you very much for your valuable comments. Upon closer inspection, we found that it does have a duplicate statement, which we have modified in line 41 to read, " Costs can also be an issue if citrus detection is performed by hyperspectral or multispectral image analysis.". Thank you for your correction.

Point 5:  Lines 58-84: what about a brief table for algorithm comparison?

Response 5:

Thank you very much for your valuable comments. In response to your question, "Make a summary table of the algorithm proposed in lines 58-84", we have presented it below using an image as well as a table.

Figure 3. One-stage object detection algorithms

In the above Figure 3, we briefly demonstrate the mainstream one-stage object detection algorithms, including SSD and YOLO series.

Table 2. References Experimental results

Authors

Citrus variety

Number of images

Image size

Main Model

mAP0.5(%)

FPS(GPU)

Params(M)

MS(MB)

Li et al. [37]

Citrus

2500

768×768

SSD

87.89

49.33

/

/

Lv et al. [38]

Citrus

620

416×416

YOLOv3

91.13

59.17

/

28.00

Mirhaji [39] et al.

Iran Dezful’s orange

766

3000×4000 and  2592×3872

YOLOv4

90.80

42.37

/

/

Chen et al. [40]

Kumquats,Nanfeng tangerines

1750

512×424

YOLOv4

96.04

16.67

/

187.00

Zheng et al. [41]

Emperorcitrus,tangerine citrus

890

4496×3000 and 2592×1944 and 3042×4032

YOLOv4

91.55

18.00

/

/

Huo et al. [42]

Shantanju

4855

1920×1080

YOLOv5s

/

12.66

/

/

Xinyang et al. [43]

Citrus

1435

640×480

YOLOv5s

99.10

187.00

/

2.80

Liu et al. [44]

Kumquat,Nanfeng tangerine,Fertile tangerine, and Shatang tangerine

1500

1280×720

YOLOv5l

98.40

52.63

50.90

/

In the Table 2, we have made a summary of the literature on one-stage object detection algorithms cited in the introduction section for the dimensions of their data sources, dataset sizes, image resolutions, models used, and model performances, and for the metrics that are not provided in the references we have filled them in the table with slashes. We hope that our answers have addressed your concerns. Thank you for your correction.

Point 6: Line 113: images have been taken at various angles: does it mean pictures have been taken even at various heights, depending on the size of the trees?

Response 6:

Thank you very much for your valuable comments. Regarding your question "does it mean pictures have been taken even at various heights, depending on the size of the trees?", we have considered this in the initial stage of collecting pictures. We have taken pictures of citrus trees of different sizes and heights, and the distance between the collection equipment and the citrus ranges from 0.3m to 2m. We made additional additions to the source of the dataset in line 113. We hope that our answers have addressed your concerns. Thank you for your correction.

Point 7: Line 147: does the area percentage of occlusion matter alone or even the shape of the occlusion in an image (i.e. a leaf in front of the fruit) can make the difference even at smaller percentages of area occlusion? This is an information that can be specified;

Response 7:

Thank you very much for your valuable comments. The percentage of occlusion here refers to when the surface area of a citrus is occluded by more than 80%, regardless of whether it is occluded by a branch, a leaf or even another citrus, and when this threshold is exceeded, we do not label the occluded citrus because its target information is severely lost. Here, we have added information in line 147 to avoid ambiguity.

Figure 4. Note on labelling

Please take a look at the Figure 4. The citrus in the cyan rectangular box has less than 80% of its surface area occluded, so we label the citrus in the box; the citrus in the yellow rectangular box has more than 80% of its surface area occluded, so we do not label the citrus; the citrus in the red rectangular box has a severe loss of silhouette information because it is in the middle of the background in the far distance, so we do not label the citrus either. We hope that our answers have addressed your concerns. Thank you for your correction.

Point 8: Line 273: the hardware resources used for the research are quite high-end material and I believe it is the right choice especially regarding GPU usage by CUDA. If authors tested other hardware configurations, they could also specify that in order to make readers understand the minimum hardware requirements to repeat similar tests on their own;

Response 8:

Thank you very much for your valuable comments. Yes, we used other hardware devices: a TITAN Xp graphics card with 12GB of video memory, a Xeon(R) E5-2680 v4 CPU, and a CUDA of 11.0, and the accuracy of the models trained on this platform did not differ from the accuracy of the models trained on the experimental platforms used in the paper, but considering the difference in training speeds, we chose the better arithmetic power of RTX A5000 for training and subsequent comparison experiments.

For your suggestion that you can specify a minimum hardware requirements, we have added after line 273: For this experiment, this paper recommends a TITAN Xp and above graphics card with 6GB of video memory and above, a CUDA version of 11.0 and above, and also a PyTorch version of no less than 1.70.

Point 9: Line 520: dataset is made of mature citrus only and future research might be dedicated to green citrus as well. If I got it correctly, you might want to assess if there is citrus in an image and if it is mature or not: if it is the purpose, authors should specify that clearly.

Response 9:

Thank you very much for your valuable comments. Regarding your query in line 520, we have 400 images of green and yellow-green citrus in the training set, and due to the sample size, we did not classify them into other categories in our study but instead labelled them as one total citrus category as well as experimented with them. If we divide three categories, i.e., green immature citrus, yellow-green semi-ripe citrus, and yellow ripe citrus, there will be a category imbalance problem. When we use a small portion of green and yellow-green citrus images for test detection, we found that the model in this paper can also detect the target correctly. However, due to the small number of samples, it lacks persuasive power, so this paper does not show it in the experimental part. As shown in the figure below, the model trained in this paper can detect green and yellow-green citrus fruits correctly.

Figure 5. Green and yellow-green citrus test results

As shown in Figure 5 above, the model in this paper was able to detect both yellow-green and green citrus, but due to quantity constraints, it is not suitable for separate illustrations. In future research, we will focus on obtaining more images of green and yellow-green citrus, forming a dataset with ripe citrus and classifying them into separate categories to assess the ripeness of the citrus while detecting the citrus in line 520. We hope that our answers have addressed your concerns. Thank you for your correction.

Reviewer 3 Report

This paper deals with Improved YOLOv7-tiny for citrus detection.

Major.

1. Why YOLOv7-tiny is chosen for detection. There are several other recently developed ML models based on YOLOv7-tiny are reported in literature. Which model has been selected for current study and what is the reason.

2. The references I mention below all use YOLOv7-tiny. Although they detect different objects, such as apples or peaches, they all use YOLOv7-tiny. To detect each object, it would be better to make a table of comparing them.

[Article Detection and Counting of Small Target Apples under Complicated Environments by Using Improved YOLOv7-tiny, Agronomy 2023, 13, 1419. https://doi.org/10.3390/agronomy13051419]

[CURI-YOLOv7: A Lightweight YOLOv7tiny Target Detector for Citrus Trees from UAV Remote Sensing Imagery Based on Embedded Device, Remote Sens. 2023, 15, 4647. https://doi.org/10.3390/rs15194647]

[YOLOv7-Peach: An Algorithm for Immature Small Yellow Peaches Detection in Complex Natural Environments, Sensors 2023, 23, 5096. https://doi.org/10.3390/s23115096]

In particular, the backbone of [https://doi.org/10.3390/agronomy13051419] uses the same backbone as this article. Of course, there are enhanced parts. It is necessary to explain how such an enhanced part was developed to detect citrus. This article jumps right into technical details without first giving an intuitive explanation of the key ideas. It is better to give more intuitive motivations to make it more accessible.

For more fluent English, it seems like a good idea to request an English-proof professional organization.

Author Response

Our response to you is in the attached word document.

--------------

Response to Reviewer 3 Comments

Thank you for your valuable comments, we have made changes and responses according to them. If we still can't answer your question, please let us know and we will be sure to correct it in time. Here is our response:

Point 1: Why YOLOv7-tiny is chosen for detection. There are several other recently developed ML models based on YOLOv7-tiny are reported in literature. Which model has been selected for current study and what is the reason.

Response 1:

Thank you very much for your valuable comments. Regarding your question, "Why YOLOv7-tiny is chosen for detection?" due to the fewer parameters and higher accuracy of YOLOv7-tiny, we chose YOLOv7-tiny as the baseline model for this study after comprehensively comparing with Faster R-CNN, YOLOv4, YOLOv5s, YOLOX-s, YOLOv7, and YOLOv7. , this paper chooses YOLOv7-tiny as the baseline model for this study. In the following, we use a short table of algorithmic experimental results to demonstrate.

Table 1. Partial model comparison experiments

Model

P(%)

R(%)

mAP0.5(%)

Params(M)

MS(MB)

Faster R-CNN

78.00

78.78

85.40

41.10

315.00

YOLOv4

92.80

88.10

94.60

24.34

245.90

YOLOv5s

92.88

90.51

94.72

7.02

13.80

YOLOX-s

89.00

86.57

91.65

8.94

102.90

YOLOv7-tiny

92.40

87.40

94.79

6.01

11.70

For the experimental results presented in the Table 1, comparing the performance of the above models in the five dimensions of P, R, mAP, Params, and Model Size, the comprehensive performance of YOLOv7-tiny is better compared to the above models, so we choose YOLOv7-tiny as our baseline model for improvement.

Regarding your question, "Which model has been chosen for the current research and why?" in the field of fruit detection, especially citrus detection algorithms, the current mainstream detection algorithms are one-stage detection algorithms, which are represented by the YOLO series, including YOLOv3, YOLOv4, YOLOv5, YOLOX, and YOLOv7, YOLOv5, YOLOX, and YOLOv7, current researchers focus on improving YOLOv4, YOLOv5, and YOLOv7 to carry out the corresponding research work. The above algorithms have a better balance between detection accuracy, algorithmic complexity, model size, detection speed, and deployment and are, therefore, loved by researchers. We hope that our answers have addressed your concerns. Thank you for your correction.

Point 2: The references I mention below all use YOLOv7-tiny. Although they detect different objects, such as apples or peaches, they all use YOLOv7-tiny. To detect each object, it would be better to make a table of comparing them.

[Article Detection and Counting of Small Target Apples under Complicated Environments by Using Improved YOLOv7-tiny, Agronomy 2023, 13, 1419. https://doi.org/10.3390/agronomy13051419]

[CURI-YOLOv7: A Lightweight YOLOv7tiny Target Detector for Citrus Trees from UAV Remote Sensing Imagery Based on Embedded Device, Remote Sens. 2023, 15, 4647. https://doi.org/10.3390/rs15194647]

[YOLOv7-Peach: An Algorithm for Immature Small Yellow Peaches Detection in Complex Natural Environments, Sensors 2023, 23, 5096. https://doi.org/10.3390/s23115096]

In particular, the backbone of [https://doi.org/10.3390/agronomy13051419] uses the same backbone as this article. Of course, there are enhanced parts. It is necessary to explain how such an enhanced part was developed to detect citrus. This article jumps right into technical details without first giving an intuitive explanation of the key ideas. It is better to give more intuitive motivations to make it more accessible.

Response 2:

Thank you very much for your valuable comments. In response to some of the references you mentioned using YOLOv7-tiny and then producing a table for comparison, we have supplemented in section 3.8 Comparison of Different Models, in addition to the results of the comparison of the explanation of the comparison of the table as shown in Table 2.

Table 2. Comparison of different models

Authors

Crop

Number of images

Image size

Models

mAP0.5(%)

FPS(GPU)

Params(M)

MS(MB)

Ma et al.[57]

Min-ne-Apple

841

320×320 410×410

YOLOv7-tiny

80.40

101.01

/

5.06

Zhang et al.[58]

Citrus

1640

4864×3648

YOLOv7-tiny

90.34

128.83

1.02

3.98

Liu et al.[59]

Yellow Peach-es

1021

4000×3000

YOLOv7

80.40

21.00

/

51.90

Our

Yongxing Bing-tang Citrus

1908

640×640

YOLOv7-tiny

96.98

169.80

2.10

4.50

Ma et al. [57] used an improved YOLOv7-tiny to detect small apple targets using the public dataset MinneApple, and compared to their model, YOLO-DCA has advantages in mAP, detection speed, number of parameters, and model size. Zhang et al. [58] used a DJI Phantom 4 RTK quadrotor UAV to collect citrus images from Sihui City, Guangdong Province, at an altitude of 50 m above the ground, and improved YOLOv7-tiny for the study. Compared with their model, YOLO-DCA is 6.64% and 40.97 higher in mAP and detection speed, respectively, while Params and Model size are only increased by 1.08M and 0.52MB. Liu et al. [59] used yellow peach images collected from the yellow peach plantation in Daping Village, Jinggangshan City, Jiangxi Province, and improved YOLOv7-tiny to conduct the study. Compared with their model, YOLO-DCA has ad-vantages regarding mAP, detection speed, and model size. Compared with the above models, YOLO-DCA balances the detection speed, model parameters, and model size while improving the accuracy.

In response to your question, "We did not present an intuitive motivation for developing these components for citrus detection and did not explain the key idea, but jumped straight into the technical details," we would like to explain the intuitive motivation, the key idea, and the technical details.

  1. In the introduction section, we describe the motivation, specifically: the improvement of our model is mainly to provide vision algorithm support for picking robots; the picking of citrus fruits in complex environments will be faced with fruit size variations, random distribution, fruit overlap, branch and leaf obstruction; secondly, the light intensity, as well as the light angle in natural environments, vary randomly, which has a more significant impact on the quality of the image; moreover, the picking robots will have a more significant influence on image quality when picking in outdoor areas; lastly, using deep convolutional neural network training models often requires more significant memory, which is not conducive to the deployment of edge devices. In addition, when the picking robot is outdoors, it will cause blurring of the image due to the varying speed of movement; finally, when using the model trained by the deep convolutional neural network, it often requires more significant memory, which is not conducive to the deployment of edge devices. To address the above existing problems, we developed the YOLOv7-tiny-based citrus detection model YOLO-DCA to provide vision algorithm support for picking robots and improve the robots' adaptability, which is our immediate motivation.
  2. In terms of key ideas, in order to be able to deploy this algorithmic model on a picking robot and for the picking robot to adapt to complex picking environments, we start from three parts:
    • reducing the model parameters and model size aspects
    • improving the feature extraction ability of the model
    • enhancing the model's ability to detect citrus at different scales

To facilitate the deployment of the model, we use DWConv to replace the ordinary convolution in ELAN to reduce the number of parameters of the model. In the neck network, the coordinate attention mechanism (CA) is combined with ordinary convolution to form CAConv to improve the feature extraction capability of the model. In the detection part, the Dynamic Head (DyHead) is used instead of the ordinary detection head to improve the model's ability to detect citrus fruits at different scales based on fusing multi-layer features.

  1. In terms of technical details, to verify the effectiveness of the improvements to YOLOv7-tiny, we conducted experiments on different components in Section 3.1 Ablation Experiments, Section 3.2 Lightweight Networks, and Section 3.3 Attention Detection Heads. The final experimental results show that our citrus detection model YOLO-DCA can maintain high accuracy while reducing the model parameters and maintaining a fast detection speed. In Sections 3.4-3.6, we design different experimental scenarios for YOLO-DCA, including occlusion, illumination, and blurriness, and the experimental results show that YOLO-DCA has better detection performance even in complex environments. In Section 3.7, we find that YOLO-DCA can identify which areas have been targeted more accurately through heat map comparison. In Section 3.8, we conduct experiments by comparing with mainstream one-stage and two-stage models, and the experimental results show that YOLO-DCA outperforms mainstream models in terms of comprehensive performance.

In addition, regarding your comment that "this paper uses the same backbone as [https://doi.org/10.3390/agronomy13051419]", we found that there is a difference between the backbone of this paper and the backbone of [https://doi.org/10.3390/agronomy13051419] after carefully reading the above literature. The above paper borrows the idea of DenseNet in the backbone network and uses jump connections for some structures. In contrast, this paper fuses with ELAN using deeply separable convolution to form ELAN-DW. Moreover, there are differences in the part of the attention to be added and the attention to be added between this paper and the above paper, as the above paper uses the ULSAM attention mechanism and adds the attention to the second downsampling part only. In contrast, we use the CA attention mechanism to add attention. We use the CA attention mechanism and combine CA with ordinary convolution, and the addition site is in the neck network. In addition, the above literature also differs from this paper in the detection head part. The above literature uses an ordinary detection head, and we use a DyHead detection head, which is a significant difference.

For the specific comparison of differences, we used a comparison diagram, as shown in Fig. The top diagram is the model structure diagram from the literature mentioned above. The bottom one is from this paper, and we used curved arrow curves to connect and compare the different parts with a short description.

Figure 1. Differences from the above literature models.

We hope that our answers have addressed your concerns. Thank you for your correction.

Round 2

Reviewer 1 Report

All the questions are answered. The manuscript can be accepted.